# Elimination of subtelomeric repeat sequences exerts little effect on telomere essential functions in *Saccharomyces cerevisiae*

Can Hu[1], Xue-Ting Zhu[1], Ming-Hong He[1], Yangyang Shao[2], Zhongjun Qin[2], Zhi-Jing Wu[1,3]*, Jin-Qiu Zhou[1,3]*

[1]The State Key Laboratory of Molecular Biology, CAS Center for Excellence in Molecular Cell Science, Shanghai Institute of Biochemistry and Cell Biology, Chinese Academy of Sciences; University of Chinese Academy of Sciences, Shanghai, China; [2]Key Laboratory of Synthetic Biology, CAS Center for Excellence in Molecular Plant Sciences, Shanghai Institute of Plant Physiology and Ecology, Chinese Academy of Sciences; University of Chinese Academy of Sciences, Shanghai, China; [3]Key Laboratory of Systems Health Science of Zhejiang Province, Hangzhou Institute for Advanced Study, University of Chinese Academy of Sciences, Hangzhou, China

*For correspondence:
zhijing.wu@utsouthwestern.edu (Z-JW);
jqzhou@sibcb.ac.cn (J-QZ)

**Abstract** Telomeres, which are chromosomal end structures, play a crucial role in maintaining genome stability and integrity in eukaryotes. In the baker's yeast *Saccharomyces cerevisiae*, the X- and Y'-elements are subtelomeric repetitive sequences found in all 32 and 17 telomeres, respectively. While the Y'-elements serve as a backup for telomere functions in cells lacking telomerase, the function of the X-elements remains unclear. This study utilized the *S. cerevisiae* strain SY12, which has three chromosomes and six telomeres, to investigate the role of X-elements (as well as Y'-elements) in telomere maintenance. Deletion of Y'-elements (SY12$^{Y\Delta}$), X-elements (SY12$^{XY\Delta+Y}$), or both X- and Y'-elements (SY12$^{XY\Delta}$) did not impact the length of the terminal TG$_{1-3}$ tracks or telomere silencing. However, inactivation of telomerase in SY12$^{Y\Delta}$, SY12$^{XY\Delta+Y}$, and SY12$^{XY\Delta}$ cells resulted in cellular senescence and the generation of survivors. These survivors either maintained their telomeres through homologous recombination-dependent TG$_{1-3}$ track elongation or underwent microhomology-mediated intra-chromosomal end-to-end joining. Our findings indicate the non-essential role of subtelomeric X- and Y'-elements in telomere regulation in both telomerase-proficient and telomerase-null cells and suggest that these elements may represent remnants of *S. cerevisiae* genome evolution. Furthermore, strains with fewer or no subtelomeric elements exhibit more concise telomere structures and offer potential models for future studies in telomere biology.

## eLife assessment

This **important** study advances our understanding of the biological significance of the DNA sequence adjacent to telomeres. The data presented **convincingly** demonstrate that subtelomeric repeats are non-essential and have a minimal, if any, role in maintaining telomere integrity of budding yeast. The work will be of interest to the telomere community specifically and the genome integrity community more broadly.

## Introduction

Telomeres, specialized nucleoprotein structures located at the end of linear chromosomes in eukaryotic cells, are crucial for maintaining genomic stability and protecting chromosomal ends from being perceived as DNA breaks (*Wellinger and Zakian, 2012*). In the budding yeast *Saccharomyces cerevisiae*, telomeric DNA consists of approximately ~300 ± 75 base pairs of $C_{1-3}A/TG_{1-3}$ repeats with a 3' G-rich single-stranded overhang (*Wellinger and Zakian, 2012*). Adjacent to the telomeric $TG_{1-3}$ repeats, there are subtelomeric repeat elements known as X- and Y'-elements, which vary between telomeres, as well as strains (*Chan and Tye, 1983a*; *Chan et al., 1983b*; *Louis, 1995*). The Y'-elements, immediately internal to the telomeric repeats, are present as a tandem array of 0–4 copies, they fall into two major size classes, 6.7 kb Y'-long (Y'-L) and 5.2 kb Y'-short (Y'-S) (*Chan and Tye, 1983a*; *Chan et al., 1983b*). Y'-elements are highly conserved with only ~2% divergence between strains (*Louis and Haber, 1992*). One entire Y'-element contains two large open-reading frames (ORFs), an ARS consensus sequence (ACS), and a STAR element (subtelomeric anti-silencing regions) consisting of binding sites for Tbf1 and Reb1 (*Chan and Tye, 1983a*; *Chan et al., 1983b*; *Fourel et al., 1999*; *Louis and Haber, 1992*). The X-element, a much more heterogeneous sequence abutting Y'-elements or telomeric repeats, contains the 473 bp 'core X' sequence and the subtelomeric repeats (STRs) A, B, C, and D (*Louis and Haber, 1991*; *Louis et al., 1994*). The STRs are found in some chromosome ends, while the 'core X' sequence is shared by all chromosomes. Recent long-read sequencing shows that subtelomeric regions display high evolutionary plasticity and are rich in various structure variants such as reciprocal translocations, transpositions, novel insertions, deletions, and duplications (*O'Donnell et al., 2023*).

Telomeric DNA elongation primarily relies on telomerase, an enzyme comprising a reverse transcriptase, an RNA component, and accessory factors (*Palm and de Lange, 2008*; *Wellinger and Zakian, 2012*). In *S. cerevisiae*, the telomerase holoenzyme consists of the reverse transcriptase Est2, the RNA template TLC1, and accessory factors Est1, Est3, Pop1/Pop6/Pop7 proteins (*Lemieux et al., 2016*; *Lendvay et al., 1996*; *Lundblad and Szostak, 1989*; *Singer and Gottschling, 1994*). In the absence of telomerase, homologous recombination can take place to replicate telomeres, resulting in telomerase-deficient 'survivors' (*Lundblad and Blackburn, 1993*; *Teng and Zakian, 1999*). These survivors are broadly categorized into Type I and Type II based on distinct telomere structures (*Lundblad and Blackburn, 1993*; *Teng and Zakian, 1999*). Type I survivors possess tandem amplified Y'-elements (both Y'-L and Y'-S) and very short $TG_{1-3}$ tracts, indicating that Y'-elements serve as substrates for homologous recombination. Type II survivors display long heterogeneous $TG_{1-3}$ tracts. On solid medium, approximately 90% of the survivors are Type I, while 10% are Type II (*Teng et al., 2000*). However, in liquid culture, Type II survivors grow faster and eventually dominate the population (*Teng and Zakian, 1999*). The proteins required for ype I and II survivor formation appear to be different. Type I survivors depend on Rad51, Rad54, Rad55, Rad57, and Pif1 (*Chen et al., 2001*; *Hu et al., 2013*; *Le et al., 1999*). while the formation of Type II survivors requires the Mre11/Rad50/Xrs2 (MRX) complex, KEOPS complex, Rad59, Sgs1, and Rad6, most of which are critical for DNA resection (*Chen et al., 2001*; *He et al., 2019*; *Hu et al., 2013*; *Johnson et al., 2001*; *Le et al., 1999*; *Huang et al., 2001*; *Nicolette et al., 2010*; *Teng et al., 2000*; *Wellinger and Zakian, 2012*; *Wu et al., 2017*). Although Type I and II pathways are working independently, Kockler et al. found that the proteins involved in each pathway can work together via two sequential steps and contribute to a unified ALT (alternative lengthening of telomeres) process (*Kockler et al., 2021*).

The amplification of Y'-elements represents a significant feature of telomere recombination in telomerase-null Type I survivors (*Lundblad and Blackburn, 1993*; *Teng and Zakian, 1999*), and as a result, extrachromosomal Y' circular DNAs have been observed in Type I survivors (*Larrivée and Wellinger, 2006*). Additionally, Y'-element acquisition has been observed in the initiation step of pre-senescence, suggesting a potential role for Y'-elements in Type II survivor formation (*Churikov et al., 2014*). Furthermore, Y'-elements are mobilized through a transposition-like RNA-mediated process involving Ty1 activity in telomerase-negative survivors (*Maxwell et al., 2004*). Y'-elements also express potential DNA helicases, Y'-Help, in telomerase-null survivors (*Yamada et al., 1998*). Thus, Y'-elements play a significant role as donors in homologous recombination-mediated telomere maintenance. The functions of X-elements, on the other hand, are less clear. The 'core X' sequence consists of an ACS element and, in most cases, an Abf1 binding site (*Louis, 1995*), and acts as a protosilencer (*Lebrun et al., 2001*). In contrast, STRs and Y'-STAR possess anti-silencing properties that limit the

spreading of heterochromatin (*Fourel et al., 1999*). Interestingly, a previous study demonstrated that telomeres with $X$-only ends (containing only X-elements) were more efficiently elongated compared to those with X-Y' ends (containing both X- and Y'-elements) in *tel1Δ rif1Δ* strains (*Craven and Petes, 1999*). Moreover, subtelomeric elements (including X-elements) and associated factors like Reb1 and Tbf1 antagonize telomere anchoring at the nuclear envelope (*Hediger et al., 2006*). However, considering that X-elements are present in all telomeres while Y'-elements are not, the specific functions of X- and Y'-elements in genome integrity after the evolution of telomerase have long been a subject of questioning (*Jäger and Philippsen, 1989*; *Zakian and Blanton, 1988*).

In wild-type yeast strain BY4742, there are 8 Y'-S and 11 Y'-L elements at the 32 telomere loci. Additionally, each telomere locus contains one X-element. The genetic deletion of all X- and Y'-elements to directly investigate the roles of X- and Y'-elements in genome integrity is a challenging and complex task. In this study, we utilized recently reported chromosome-fused budding yeast strains (*Shao et al., 2018*) to eliminate both X- and Y'-elements completely. This approach allows us to reinvestigate the roles of X- and Y'-elements at telomeres.

## Results

### Telomere recombination in telomerase-null chromosome-fused yeast strains SY1 to SY12

The functions of Y'-elements have been previously linked to telomere recombination (*Churikov et al., 2014*; *Larrivée and Wellinger, 2006*; *Lundblad and Blackburn, 1993*; *Teng and Zakian, 1999*). To further investigate the role of Y'-elements in telomere recombination, we utilized a series of chromosome-fused budding yeast strains derived from the wild-type BY4742 strain, including SY1, SY3, SY5, SY7, SY8, SY9, SY10, SY11, SY12, and SY13 (also referred to as SYn for convenience) (*Figure 1A*; *Shao et al., 2018*). The remaining subtelomeric elements in SY8 to SY13 strains are listed in *Supplementary file 2*. We excluded SY14 from these experiments since the presence of circular chromosome was prominent in SY14 *tlc1Δ* cells (one fused chromosome) (*Wu et al., 2020*), We generated haploid SYn *tlc1Δ TLC1* strains by deleting the chromosomal copy of the *TLC1* gene and introducing a plasmid-borne wild-type *TLC1* gene (pRS316-*TLC1*). Clones that lost the pRS316-*TLC1* plasmid (containing the *URA3* marker) were identified upon counter-selection on 5'-fluoroorotic-acid (5'-FOA) plates and were subsequently re-streaked on YPD plates for at least nine cycles for survivor formation (referred to as the 'multiple-colony streaking assay' in 'Materials and methods'). The telomere patterns of the survivors were then determined through Southern blotting assay (*Figure 1B–D*).

The canonical telomerase-independent survivors can be broadly categorized into two types: Type I and Type II survivors, based on the restriction fragments generated after XhoI digestion (*Lundblad and Blackburn, 1993*; *Teng and Zakian, 1999*). Type I survivors exhibit tandem duplication of Y'-elements and very short $TG_{1-3}$ tracts, while Type II survivors contain long heterogeneous $TG_{1-3}$ sequences. Consistent with previous reports, BY4742 *tlc1Δ* cells generated both Type I (subtelomeric Y'-element recombination) and Type II ($TG_{1-3}$ recombination) survivors (*Figure 1B*; *Hu et al., 2013*). Intriguingly, as the number of chromosomes decreased, the frequency of Type II survivors gradually diminished, while Type I survivors became the predominant type (*Figure 1B–D*). Furthermore, non-canonical survivors with distinct patterns from Type I or Type II emerged in SY9 *tlc1Δ* (six chromosomes), SY10 *tlc1Δ* (five chromosomes), SY11 *tlc1Δ* (four chromosomes), SY12 *tlc1Δ* (three chromosomes), and SY13 *tlc1Δ* (two chromosomes) (*Figure 1C and D* indicated by triangles at the bottom of the panels). Notably, the Y'-telomere band of ~1.2 kb was not detected in two clones of SY11 *tlc1Δ* cells (clones 2 and 5), the majority of clones of SY12 *tlc1Δ* cells (except for clones 9, 14, and 15), and the majority of clones of SY13 *tlc1Δ* cells (except for clones 1, 4, 8, and 10) (*Figure 1D*). We speculate that either the Y'-elements have eroded or the chromosomal ends containing Y'-elements have fused with other ends in these non-canonical survivors. These findings suggest that the ratio of survivor types is influenced by the number of chromosomes.

### Characterizing the survivor pattern in SY12

To determine the chromosomal end structures of the non-canonical survivors shown in *Figure 1*, we selected SY12 *tlc1Δ* survivors for further analysis. In the SY12 strain, there are six telomeres corresponding to the native chromosomes I-L, X-R, XIII-L, XI-R, XVI-L, and XIV-R. We employed Southern

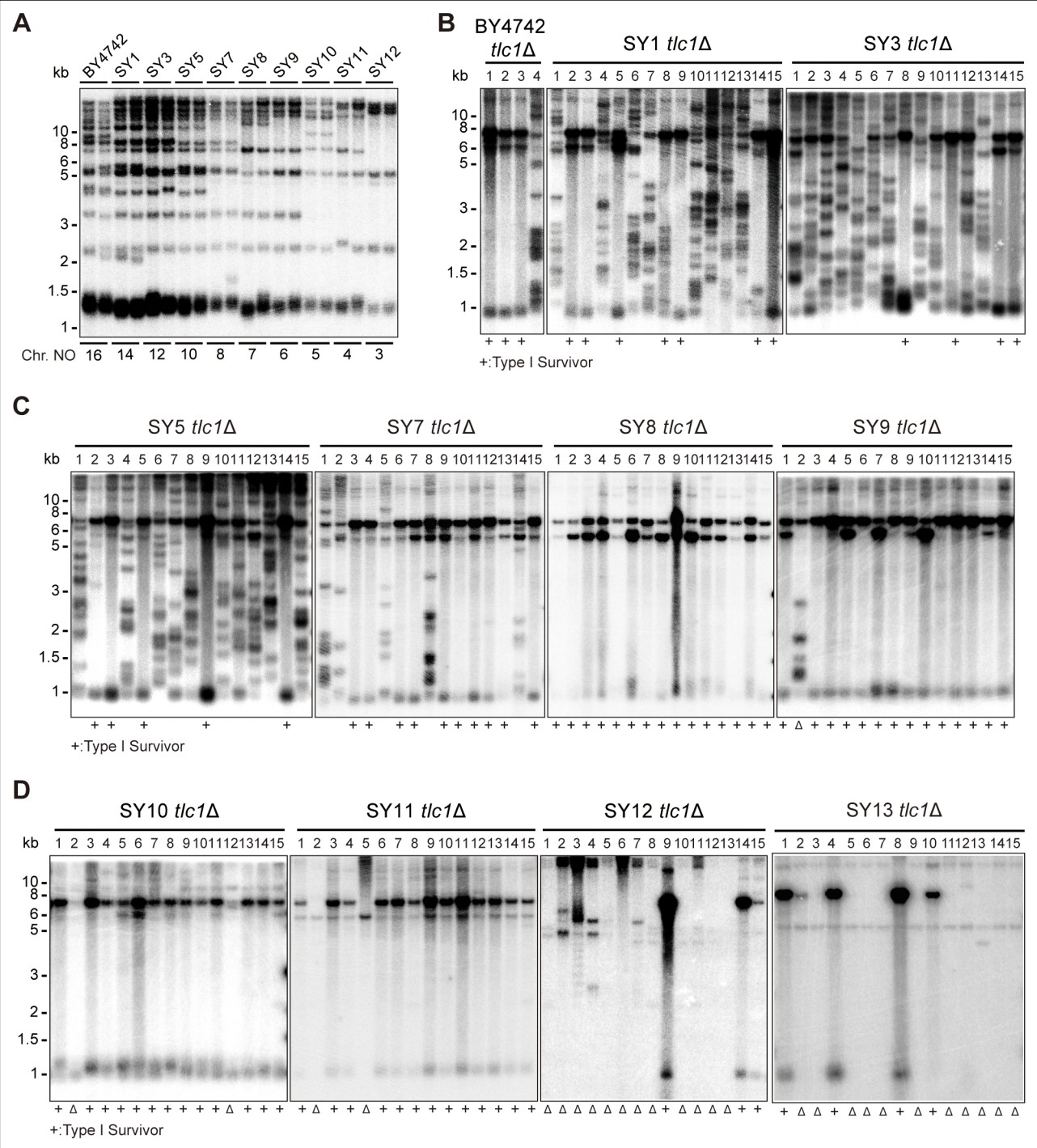

**Figure 1.** Telomere structures in SYn *tlc1Δ* survivors. Telomere Southern blotting assay was performed to examine telomere structure. The genomic DNA extracted from BY4742 (wild type) and SYn strains (labeled on top) was digested with XhoI and subjected to Southern hybridization with a TG$_{1-3}$ probe. (**A**) Telomerase-proficient strains (labeled on top), whose chromosome numbers are labeled at the bottom. Two independent clones of each strain were examined. (**B–D**) SYn *tlc1Δ* survivors generated on plates. In total, 4 (BY4742 *tlc1Δ*) and 15 (SYn *tlc1Δ*) individual survivor clones (labeled on top of each panel) of each strain were examined. '+' at the bottom indicates Type I survivors. 'Δ' marks the survivors which are non-canonical Type I or Type II.

The online version of this article includes the following source data for figure 1:

**Source data 1.** Original file for the Southern blotting analysis in *Figure 1A*.

**Source data 2.** File containing *Figure 1A* and original scans of the relevant Southern blotting analysis.

*Figure 1 continued on next page*

*Figure 1 continued*

**Source data 3.** Original file for the Southern blotting analysis in *Figure 1B* for BY4742 *tlc1Δ* and SY1 *tlc1Δ*.

**Source data 4.** Original file for the Southern blotting analysis in *Figure 1B* for SY3 *tlc1Δ*.

**Source data 5.** File containing *Figure 1B* and original scans of the relevant Southern blotting analysis.

**Source data 6.** Original file for the Southern blotting analysis in *Figure 1C* for SY5 *tlc1Δ*.

**Source data 7.** Original file for the Southern blotting analysis in *Figure 1C* for SY7 *tlc1Δ* and SY8 *tlc1Δ*.

**Source data 8.** Original file for the Southern blotting analysis in *Figure 1C* for SY9 *tlc1Δ*.

**Source data 9.** File containing *Figure 1C* and original scans of the relevant Southern blotting analysis.

**Source data 10.** Original file for the Southern blotting analysis in *Figure 1D* for SY10 *tlc1Δ*.

**Source data 11.** Original file for the Southern blotting analysis in *Figure 1D* for SY11 *tlc1Δ*.

**Source data 12.** Original file for the Southern blotting analysis in *Figure 1D* for SY12 *tlc1Δ*.

**Source data 13.** Original file for the Southern blotting analysis in *Figure 1D* for SY13 *tlc1Δ*.

**Source data 14.** File containing *Figure 1D* and original scans of the relevant Southern blotting analysis.

blotting after NdeI digestion to validate the telomere and subtelomere structures (*Figure 2—figure supplement 1A*). The results revealed that, in the SY12 strain used in our study, only the XVI-L telomere contained a single copy of the Y'-element, while all telomeres harbored X-elements (*Figure 2—figure supplement 1B*). For simplicity, we referred to the chromosomes containing the original I, XIII, and XVI as chromosome 1, 2, and 3, respectively (*Figure 2A*, left panel).

We conducted a re-examination of telomere recombination upon telomerase inactivation in SY12 cells. Deletion of *TLC1* in SY12 cells resulted in cell senescence, and different clones recovered at various time points in liquid medium (*Figure 2B*). Telomere Southern blotting analysis showed progressive shrinking of the telomeric XhoI fragments over time, and TG$_{1-3}$ recombination occurred to maintain telomeres (*Figure 2C*). Since the liquid culture contained a mixture of different colonies, we employed a multiple-colony streaking assay and Southern blotting analysis to examine the telomere patterns of 50 independent SY12 *tlc1Δ* survivors (*Figure 2D*, *Figure 2—figure supplement 2*). Among these survivors, eight clones (labeled in red, 16% of the survivors tested) exhibited the typical Type I telomere structure characterized by Y'-element amplification (*Figure 2D and E* and *Supplementary file 5*). This was confirmed by Southern blotting analysis using a Y' probe (*Figure 2—figure supplement 2*). The emergence of Type I survivor in SY12 strain which only contain one Y'-element indicates that multiple Y'-elements in tandem are not strictly required for Type I formation. Clone 1 (labeled in orange, 2% of the survivors tested) displayed heterogeneous telomeric TG$_{1-3}$ tracts (*Figure 2D and E* and *Supplementary file 5*), indicating it was a Type II survivor. This was further confirmed by restoring the telomere length to the level observed in SY12 cells through the reintroduction of the *TLC1* gene into one representative clone (named SY12 *tlc1Δ*-T1) and subsequent passaging on yeast complete (YC) medium lacking uracil (Ura-) for 20 cycles (*Figure 2—figure supplement 3A*).

Notably, 10 of the examined clones (labeled in blue, 20% of the survivors tested) displayed no telomere signals associated with canonical Type I or II survivors (*Figure 2D and E* and *Supplementary file 5*). Their hybridization patterns were strikingly similar to those of SY14 *tlc1Δ* survivors (*Wu et al., 2020*), which survived through intra-chromosomal circularization. To investigate whether the three chromosomes in these SY12 *tlc1Δ* survivors had undergone intra-chromosomal fusions, we selected a clone, namely SY12 *tlc1Δ*-C1, and performed PCR-mapping assay to determine the erosion points of each chromosome end, as previously described (*Wu et al., 2020*). A PCR product of the predicted length would be obtained only if the corresponding chromosome region was intact. The PCR-mapping assay precisely identified the borders of telomere erosion for the three chromosomes in SY12 *tlc1Δ*-C1 cells. For chromosome 1 (*Figure 2A*, left panel), the chromosome regions approximately 3.3 kb and 1.9 kb proximal to telomere I-L and X-R, respectively, had been lost (*Figure 2—figure supplement 4* and *Supplementary file 3*). Regarding chromosome 2, the terminal ~3.8 kb of telomere XIII-L and ~2.5 kb of telomere XI-R remained intact (*Figure 2—figure supplement 4* and *Supplementary file 3*). For chromosome 3, the terminal ~0.1 kb of telomere XVI-L was intact, while the terminal ~3.4 kb of telomere XIV-R was preserved (*Figure 2—figure supplement 4* and *Supplementary file 3*). To confirm the chromosome fusion events, we performed PCR-sequencing analysis. If a given pair of primers, oriented to different chromosome ends, produced PCR products, it indicated that the corresponding

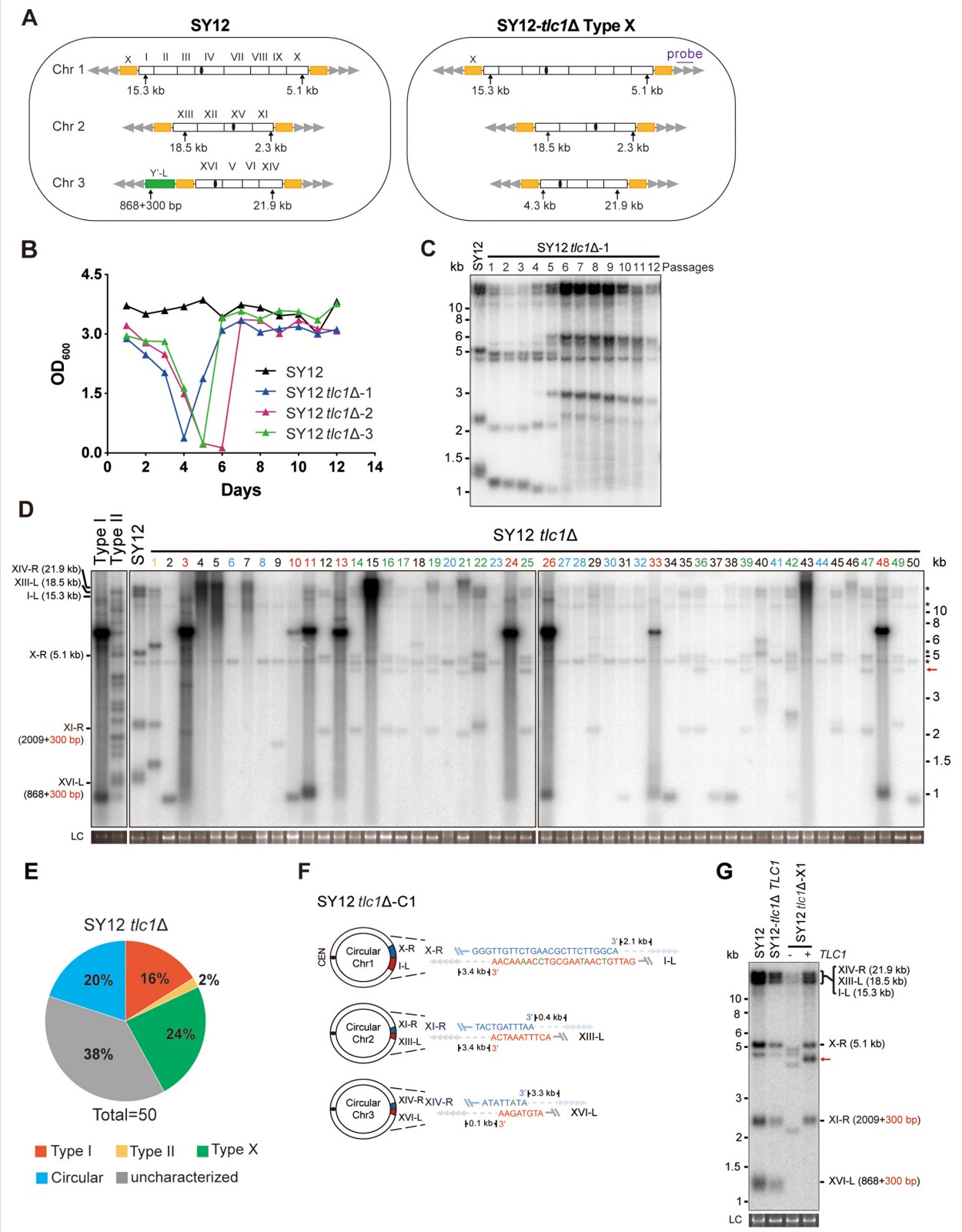

**Figure 2.** Survivor formation in SY12 *tlc1Δ* strain. (**A**) Schematic representation of chromosome (and telomere) structures (not drawn to scale) in the SY12 strain (left panel) and the Type X survivor (right panel). The Roman numerals, native chromosomes; the Arabic numerals on the left, chromosome numbers of SY12; yellow box, X-element; green box, Y′-element; tandem gray triangles, telomeres; black circles, centromere; vertical arrows and numbers, positions and lengths of the terminal XhoI digestion fragments detected by the telomeric $TG_{1-3}$ probe. Chromosome numbers are omitted in

*Figure 2 continued on next page*

*Figure 2 continued*

the Type X survivor (right panel). (**B**) Cell viability assay in liquid medium. The growth of SY12 (labeled in black) and SY12 *tlc1Δ* (three clones labeled in blue, purple, and green, respectively) strains were monitored every 24 hr for 12 d. (**C**) Telomeric Southern blotting assay of SY12 *tlc1Δ* survivors. Genomic DNAs prepared from SY12 *tlc1Δ* survivors assayed in (**B**) were digested with XhoI and subjected to Southern blotting with a TG$_{1-3}$ probe. (**D**) Telomere Southern blotting assay of SY12 *tlc1Δ* survivors obtained on solid medium. Genomic DNA from 50 independent SY12 *tlc1Δ* clones (labeled on top) was digested with XhoI and hybridized to a telomere-specific TG$_{1-3}$ probe. Type II survivors: in orange; Type I survivors: in red; circular survivors: in blue; Type X survivors: in green; uncharacterized survivors: in black. Theoretical telomere restriction fragments of the SY12 strain are indicated on the left. The red arrows indicate the new band of about 4.3 kb emerged in Type X survivors. The asterisks indicate the non-specific bands. Genomic DNA stained with Gelred was used as a relative loading control (LC). (**E**) The ratio of survivor types in SY12 *tlc1Δ* strain. n = 50; Type I, in red; Type II, in orange; Type X, in green; uncharacterized survivor, in gray; circular survivor, in blue. (**F**) Schematic of three circular chromosomes and fusion sequences in the SY12 *tlc1Δ*-C1 survivor. The sequence in blue indicates the sequences of X-R, XI-R, or XIV-R, the sequence in red indicates the sequences of I-L, XIII-L, or XVI-L. Bases in green are mis-paired. The numbers above or below the schematic line (chromosome) indicate the distance to the corresponding telomeres. (**G**) Telomere Southern blotting analysis of an SY12 *tlc1Δ* Type X survivor at the 20th re-streak after *TLC1* reintroduction. The red arrows indicate the new band of about 4.3 kb emerged in Type X survivors. LC: loading control.

The online version of this article includes the following source data and figure supplement(s) for figure 2:

**Source data 1.** File containing output results of growth analysis of the SY12 *tlc1Δ* strain in *Figure 2B*.

**Source data 2.** Original file for the Southern blotting analysis in *Figure 2C*.

**Source data 3.** File containing *Figure 2C* and original scans of the relevant Southern blotting analysis.

**Source data 4.** Original file for the Southern blotting analysis in *Figure 2D*.

**Source data 5.** Original file for the Southern blotting analysis in *Figure 2D*.

**Source data 6.** Original file for the loading control of Southern blotting analysis in *Figure 2D*.

**Source data 7.** Original file for the loading control of Southern blotting analysis in *Figure 2D*.

**Source data 8.** File containing *Figure 2D* and original scans of the relevant Southern blotting analysis.

**Source data 9.** File containing the original scans of the loading control of the Southern blotting analysis in *Figure 2D*.

**Source data 10.** Original file for the Southern blotting analysis in *Figure 2G*.

**Source data 11.** Original file for the loading control of Southern blotting analysis in *Figure 2G*.

**Source data 12.** File containing *Figure 2G* and original scans of the relevant Southern blotting analysis.

**Source data 13.** File containing the original scans of the loading control of the Southern blotting analysis in *Figure 2G*.

**Figure supplement 1.** Characterization of SY12 strain.

**Figure supplement 1—source data 1.** Original file for the Southern blotting analysis in *Figure 2—figure supplement 1B*.

**Figure supplement 1—source data 2.** Original file for the Southern blotting analysis in *Figure 2—figure supplement 1B*.

**Figure supplement 1—source data 3.** File containing *Figure 2—figure supplement 1B* and original scans of the relevant Southern blotting analysis.

**Figure supplement 2.** Telomere Southern blot with a Y'-element probe examining SY12 *tlc1Δ* survivors.

**Figure supplement 2—source data 1.** Original file for the Southern blotting analysis in *Figure 2—figure supplement 2*.

**Figure supplement 2—source data 2.** Original file for the Southern blotting analysis in *Figure 2—figure supplement 2*.

**Figure supplement 2—source data 3.** File containing *Figure 2—figure supplement 2* and original scans of the relevant Southern blotting analysis.

**Figure supplement 3.** Southern blotting results of reintroduce *TLC1* into SY12 *tlc1Δ* survivors.

**Figure supplement 3—source data 1.** Original file for the Southern blotting analysis in *Figure 2—figure supplement 3A*.

**Figure supplement 3—source data 2.** Original file for the Southern blotting analysis in *Figure 2—figure supplement 3B*.

**Figure supplement 3—source data 3.** Original file for the loading control of the Southern blotting analysis in *Figure 2—figure supplement 3A*.

**Figure supplement 3—source data 4.** Original file for the loading control of the Southern blotting analysis in *Figure 2—figure supplement 3B*.

**Figure supplement 3—source data 5.** File containing *Figure 2—figure supplement 3A and B* and original scans of the relevant Southern blotting analysis.

**Figure supplement 3—source data 6.** File containing the loading control of the relevant Southern blotting analysis in *Figure 2—figure supplement 3A and B*.

**Figure supplement 4.** Borders of erosion of the SY12 *tlc1Δ*-C1 survivor are defined by PCR mapping.

**Figure supplement 5.** Pulsed-field gel electrophoresis (PFGE) result of circular survivors.

**Figure supplement 5—source data 1.** Original file for the PFGE analysis in *Figure 2—figure supplement 5*.

**Figure supplement 5—source data 2.** File containing *Figure 2—figure supplement 5* and original scans of the relevant PFGE analysis.

**Figure supplement 6.** Telomere structure determination of type X survivor.

*Figure 2 continued on next page*

*Figure 2 continued*

**Figure supplement 7.** Survivor formation in SY12 *tlc1Δ rad52Δ* strain.

**Figure supplement 7—source data 1.** File containing output results of growth analysis of the SY12 *tlc1Δ rad52Δ* strain in *Figure 2—figure supplement 7A*.

**Figure supplement 7—source data 2.** Original file for the Southern blotting analysis of the SY12 *tlc1Δ rad52Δ* strain in *Figure 2—figure supplement 7B*.

**Figure supplement 7—source data 3.** Original file for the loading control of the Southern blotting analysis of the SY12 *tlc1Δ rad52Δ* strain in *Figure 2—figure supplement 7B*.

**Figure supplement 7—source data 4.** File containing *Figure 2—figure supplement 7B* and original scans of the relevant Southern blotting analysis.

**Figure supplement 7—source data 5.** File containing the loading control of the relevant Southern blotting analysis in *Figure 2—figure supplement 7B*.

**Figure supplement 8.** Southern blotting result of SY12 *tlc1Δ rad51Δ* and SY12 *tlc1Δ rad50Δ* survivors.

**Figure supplement 8—source data 1.** Original file for the Southern blotting analysis of the SY12 *tlc1Δ rad51Δ* strain in *Figure 2—figure supplement 8A*.

**Figure supplement 8—source data 2.** Original file for the loading control of the Southern blotting analysis of the SY12 *tlc1Δ rad51Δ* strain in *Figure 2—figure supplement 8A*.

**Figure supplement 8—source data 3.** File containing the loading control of the relevant Southern blotting analysis of the SY12 *tlc1Δ rad51Δ* strain in *Figure 2—figure supplement 8A*.

**Figure supplement 8—source data 4.** Original file for the Southern blotting analysis of the SY12 *tlc1Δ rad50Δ* strain in *Figure 2—figure supplement 8B*.

**Figure supplement 8—source data 5.** File containing *Figure 2—figure supplement 8A and B* and original scans of the relevant Southern blotting analysis.

arms had fused. The results revealed that the three chromosomes in SY12 *tlc1Δ*-C1 cells had undergone intra-chromosomal fusions through microhomology-mediated end joining (MMEJ) (*Wu et al., 2020*), resulting in the formation of circular chromosomes (*Figure 2F* and *Supplementary file 3*). Notably, the fusion junctions of the three chromosomes in SY12 *tlc1Δ*-C1 cells differed in nucleotide sequence and length (22 bp, 8 bp, and 5 bp in chromosomes 1, 2, and 3, respectively). Moreover, the sequences involved in the ends-fusion were not perfectly complementary (*Figure 2F*). For example, the fusion sequence of chromosome 3 was 5 bp long and contained one mismatch. To further verify the chromosome structure in the 'circular survivors' SY12 *tlc1Δ*-C1 (*Figure 2F*), we performed the pulsed-field gel electrophoresis (PFGE) analysis. Control strains included SY12 (three linear chromosomes) and SY15 (one circular chromosome). The PFGE result confirmed that like the single circular chromosome in SY15 cells, the circular chromosome in the SY12 *tlc1Δ*-C1 survivors could not enter the gel, while the linear chromosomes in SY12 were separated into distinct bands, as expected (*Figure 2—figure supplement 5*). Thus, the survivors shown in *Figure 2D*, which displayed an identical hybridization pattern to the SY12 *tlc1Δ*-C1 clone, were all likely 'circular survivors'. Consistently, the telomere signals detected in the SY12 strain were still not observed in the SY12 *tlc1Δ*-C1 survivor after reintroducing a plasmid-borne wild-type *TLC1* gene (*Figure 2—figure supplement 3B*).

Twelve clones of SY12 *tlc1Δ* survivors (labeled in green, 24% of the survivors tested) exhibited no Y'-telomere signals compared to SY12 cells but displayed different lengths of $TG_{1-3}$ tracts (*Figure 2D and E* and *Supplementary file 5*). Due to their non-canonical telomere structures, characterized by the absence of both Y'- amplification and superlong $TG_{1-3}$ sequences, we designated these SY12 *tlc1Δ* survivors (labeled in green, *Figure 2D*) as Type X. In Type X survivors, the DNA bands with sizes of approximately 2.3 kb, 5.1 kb, 15.3 kb, 18.5 kb, and 21.9 kb were roughly comparable to the telomeres of XI-R, X-R, I-L, XIII-L, and XIV-R in SY12 cells (indicated on the left in the panel). The newly emerged band at approximately 4.3 kb likely originated from the XVI-L telomere (indicated by the red arrow on the right in the panel) (*Figure 2D*), where the Y'-elements had been eroded, leaving only the $TG_{1-3}$ tracts at the very ends (*Figure 2A*, right panel). It remains unclear whether Y'-element erosion is common in telomerase-null BY4742 Type II survivors. However, in SY12 *tlc1Δ* cells, the remaining single copy of the Y'-element could not find homology sequences to repair telomeres, whereas the multicopy X-element could easily find homology sequences to repair telomeres and form the Type X survivors. To verify this notion, we reintroduced the *TLC1* gene into one representative clone (named SY12 *tlc1Δ*-X1) and examined the telomere length. As expected, the telomeres of X-R and XI-R were

restored to the lengths observed in wild-type SY12 cells, and accordingly, the newly emerged 4.3 kb band was also elongated (*Figure 2G*). Given that the restriction fragments of telomeres I-L (15.3 kb), XIII-L (18.5 kb), and XIV-R (21.9 kb) were quite long, detecting minor changes in telomere length was challenging under the assay conditions of Southern blotting. To determine the chromosomal end structure of the Type X survivor, we randomly selected a typical Type X survivor, and performed PCR-sequencing analysis. The results revealed the intact chromosome ends for I-L, X-R, XIII-L, XI-R, and XIV-R, albeit with some mismatches compared with the *S. cerevisiae* S288C genome (http://www.yeastgenome.org/), which possibly arising from recombination events that occurred during survivor formation. Notably, the sequence of the Y'-element in XVI-L could not be detected, while the X-element remained intact (*Figure 2—figure supplement 6*). These data indicated that Type X survivors possess linear chromosomes with telomeres terminating in $TG_{1-3}$ repeats, while the Y'-element has been eroded (*Figure 2A*, right panel). Consistently, no Y' signals were detected in these 12 Type X survivors (labeled in green, *Figure 2—figure supplement 2*), suggesting that the Y'-element has not been translocated to other telomeres and is not essential for yeast cell viability.

In addition to the aforementioned Type I, Type II, circular, and Type X survivors, there were some clones (labeled in black, 38% of the survivors tested) which exhibited non-uniform telomere patterns and were not characterized (*Figure 2D and E* and *Supplementary file 5*). We speculated that combinations of diverse mechanisms were occurring within each 'uncharacterized survivor'. For instance, in the case of two survivors (clones 9 and 18, *Figure 2D*) in which only one hybridization signal could be detected, pointing to the possibility that two chromosomes underwent intra-chromosomal fusions while one retained its ends through $TG_{1-3}$ recombination. However, the sizes of the two telomere restriction fragments on the linear chromosome were too close to be distinguished and separated, resulting in only one hybridization signal. Alternatively, it is also plausible that three chromosomes experienced intra-chromosomal fusions, with one fusion point containing $TG_{1-3}$ repeats. For the uncharacterized clones 4, 5, 7, 15, and 43, they exhibited significant amplification of $TG_{1-3}$ sequences, and the telomeres of these survivors did not resolve into distinct bands (*Figure 2D*). We hypothesize that the observed telomere patterns in these survivors could be attributed to extensive $TG_{1-3}$ recombination. However, we cannot exclude the possibility of coexisting diverse mechanisms within a survivor, such as telomere elongation through $TG_{1-3}$ amplification, as well as intra- and inter-chromosomal fusions. Since we could not figure out the telomere structures in these survivors, we classified them as 'uncharacterized survivors'.

To further determine the genetic requirements for survivors in SY12, we constructed the SY12 *tlc1Δ rad52Δ* pRS316-*TLC1* strain. The plasmid-borne wild-type *TLC1* gene (pRS316-*TLC1*) was counter-selected on 5'-FOA plates. SY12 *tlc1Δ rad52Δ* cells were measured by the cell viability assay (see 'Materials and methods'). The results showed double deletion of *TLC1* and *RAD52* in SY12 strain could slightly accelerate senescence, and SY12 *tlc1Δ rad52Δ* survivors could be generated but took much longer to recover than the SY12 *tlc1Δ* survivors (*Figure 2—figure supplement 7A*), suggesting that Rad52 is not strictly required for survivor generation in the SY12 strain in liquid. We also passaged SY12 *tlc1Δ rad52Δ* cells on solid medium until survivor emerged. Southern blotting of 25 clones revealed that neither Type I nor II survivors were found, and instead circular survivors except clone 20 were obtained (labeled in blue, *Figure 2—figure supplement 7B*). We conclude that the formation of circular survivors in the SY12 *tlc1Δ rad52Δ* strain is mediated by MMEJ as observed in the SY14 *tlc1Δ rad52Δ* strain (*Wu et al., 2020*), but not *RAD52* mediate pathways. Since no Type X survivor was detected in SY12 *tlc1Δ rad52Δ* strain, we constructed the SY12 *tlc1Δ rad51Δ* pRS316-*TLC1* and SY12 *tlc1Δ rad50Δ* pRS316-*TLC1* strain to investigate on which pathway the formation of the Type X survivor relied. After being counter-selected on 5'-FOA plates, cells were passaged on solid medium until survivor arose. Southern blotting assay indicated the emergence of Type X survivors even in the absence of Rad51 (labeled in green, clones 2, 5, 11, and 18, *Figure 2—figure supplement 8A*). In contrast, no Type X survivor was detected in the SY12 *tlc1Δ rad50Δ* strain (*Figure 2—figure supplement 8B*). These data suggest that the formation of the Type X survivor depends on Rad50-mediated Type II pathway.

Taken together, our results indicate that telomerase inactivation in SY12 cells leads to cell senescence and the emergence of survivors with diverse telomere patterns, including Y'-amplification (Type I), elongated $TG_{1-3}$ tracts (Type II), intra-chromosomal end-to-end joining (circular), Y'- loss (Type X), and uncharacterized.

## Deletion of all of the X- and Y'-elements in the SY12 strain

We aimed to determine whether the subtelomeric X-elements are dispensable or not. In the SY12 strain, there are six X-elements distributed among six telomeres (*Figure 2A*, left panel). To precisely delete all X- and Y'-elements in SY12 strains, we employed a method that combines the efficient CRISPR-Cas9 cleavage system with the robust homologous recombination activity of yeast, as previously described (*Shao et al., 2018*; *Shao et al., 2019*). Briefly, the Cas9 nuclease cleaved the unique DNA sequences adjacent to the subtelomeric region (site S1) with the guidance of gRNA1. The resulting chromosome break was repaired through homologous recombination (HR) using the provided chromosome ends excluding the X- and Y'-elements. Subsequently, the *URA3* marker and the guide RNA expression plasmid (pgRNA) were eliminated by inducing gRNA2 expression on pCas9 using galactose (*Figure 3—figure supplement 1*). This approach allowed us to initially delete the Y'-element and X-element in XVI-L, generating the SY12$^{Y\Delta}$ strain (*Figure 3A*, *Supplementary file 4*, and *Supplementary file 6*). Subsequently, through five successive rounds of deletions, we removed all remaining X-elements, resulting in the SY12$^{XY\Delta}$ strain (*Figure 3A*, *Supplementary file 4*, and *Supplementary file 6*). To confirm the series of deletions, we performed PCR analysis using a primer located within the deletion region and another primer annealing upstream of the region (indicated by purple arrows in *Figure 3—figure supplement 1*, primers are shown in *Supplementary file 1*). This analysis verified the complete deletion of the subtelomeric X- and Y'-elements (*Figure 3B*, rows 3–7). Additionally, we conducted a separate PCR analysis using primers specific to either X- or Y'-elements, which confirmed the absence of both X- and Y'-elements in the SY12$^{XY\Delta}$ strain (*Figure 3B*, row 8). Subsequently, we inserted a Y'-long element (cloned from the native XVI-L sequence, which does not contain the centromere-proximal short telomere sequence) into the left arm of chromosome 3 in the SY12$^{XY\Delta}$ strain, resulting in the SY12$^{XY\Delta+Y}$ strain containing a single Y'-element but no X-element (*Figure 3A* and *Supplementary file 4*). The successful insertion was confirmed by PCR analysis (*Figure 3B*, lane 9).

## Subtelomeric X- and Y'-elements are dispensable for cell proliferation, various stress responses, telomere length control, and telomere silencing

The SY12$^{Y\Delta}$, SY12$^{XY\Delta}$, and SY12$^{XY\Delta+Y}$ cells, cultured in YPD medium at 30°C, exhibited the same cell morphology as the parental strains SY12 and BY4742 (*Figure 3C*). To assess the stability of their genomes, we restreaked several clones of SY12$^{Y\Delta}$, SY12$^{XY\Delta}$, and SY12$^{XY\Delta+Y}$ strains on YPD plates for a total of 61 times at 2-day intervals (*Figure 3D*). Similar to the SY12 strain, the progeny colonies of SY12$^{Y\Delta}$, SY12$^{XY\Delta}$, and SY12$^{XY\Delta+Y}$ grew robustly on solid medium (*Figure 3D*). Moreover, SY12$^{Y\Delta}$, SY12$^{XY\Delta}$, and SY12$^{XY\Delta+Y}$ cells exhibited growth rates comparable to those of SY12 and BY4742 cells in liquid medium (*Figure 3E*). Fluorescence-activated cell sorting (FACS) analysis revealed that SY12$^{Y\Delta}$, SY12$^{XY\Delta}$, and SY12$^{XY\Delta+Y}$ had the same 1C and 2C DNA content as wild-type cells (*Figure 3F*), indicating that the X- and Y'-elements are not necessary for cell proliferation under normal conditions. Additionally, the growth of SY12$^{Y\Delta}$, SY12$^{XY\Delta}$, and SY12$^{XY\Delta+Y}$ cells at different temperatures (24 and 37°C) (*Figure 3G*, upper panel) closely resembled that of SY12 and BY4742 cells. Furthermore, SY12$^{Y\Delta}$, SY12$^{XY\Delta}$, SY12$^{XY\Delta+Y}$, SY12, and BY4742 cells exhibited similar sensitivities to various genotoxic agents, including hydroxyurea (HU), camptothecin (CPT), and methyl methanesulfonate (MMS) (*Figure 3G*, lower panel). These results indicate that the X- and Y'-elements are dispensable for cellular responses to cold or heat treatment and DNA damage challenges, consistent with a recent study of 'synthetic yeast genome project', namely Sc2.0, showing that thousands of genome-wide edits, including the deletion of subtelomeric repetitive sequences, deletion of introns, and relocation of tRNAs genes, yielded a strain that displays comparable growth with wild-type strain (*Richardson et al., 2017*; *Zhao et al., 2023*).

Next, we examined the effects of X- and Y'-element elimination on telomeres. Southern blotting assay revealed that SY12$^{Y\Delta}$, SY12$^{XY\Delta}$, and SY12$^{XY\Delta+Y}$ cells maintained stable telomeres at a length of approximately 300 bp, comparable to that in SY12 cells (*Figure 4A*), indicating that the X- and Y'-elements are not required for telomere length regulation. To determine whether the deletion of X- and Y'-elements abolishes telomere silencing, we constructed haploid strains of SY12$^{Y\Delta}$ *sir2Δ*, SY12$^{XY\Delta}$ *sir2Δ*, SY12$^{XY\Delta+Y}$ *sir2Δ*, SY12 *sir2Δ*, and BY4742 *sir2Δ*. We then performed real-time RT-PCR to quantify the expression of the *MPH3* and *HSP32* genes, located near the subtelomeric region of X-R (X-only end) and XVI-L (X-Y' end), respectively (*Figure 4B*), and found that the increase of the *MPH3* or *HSP32*

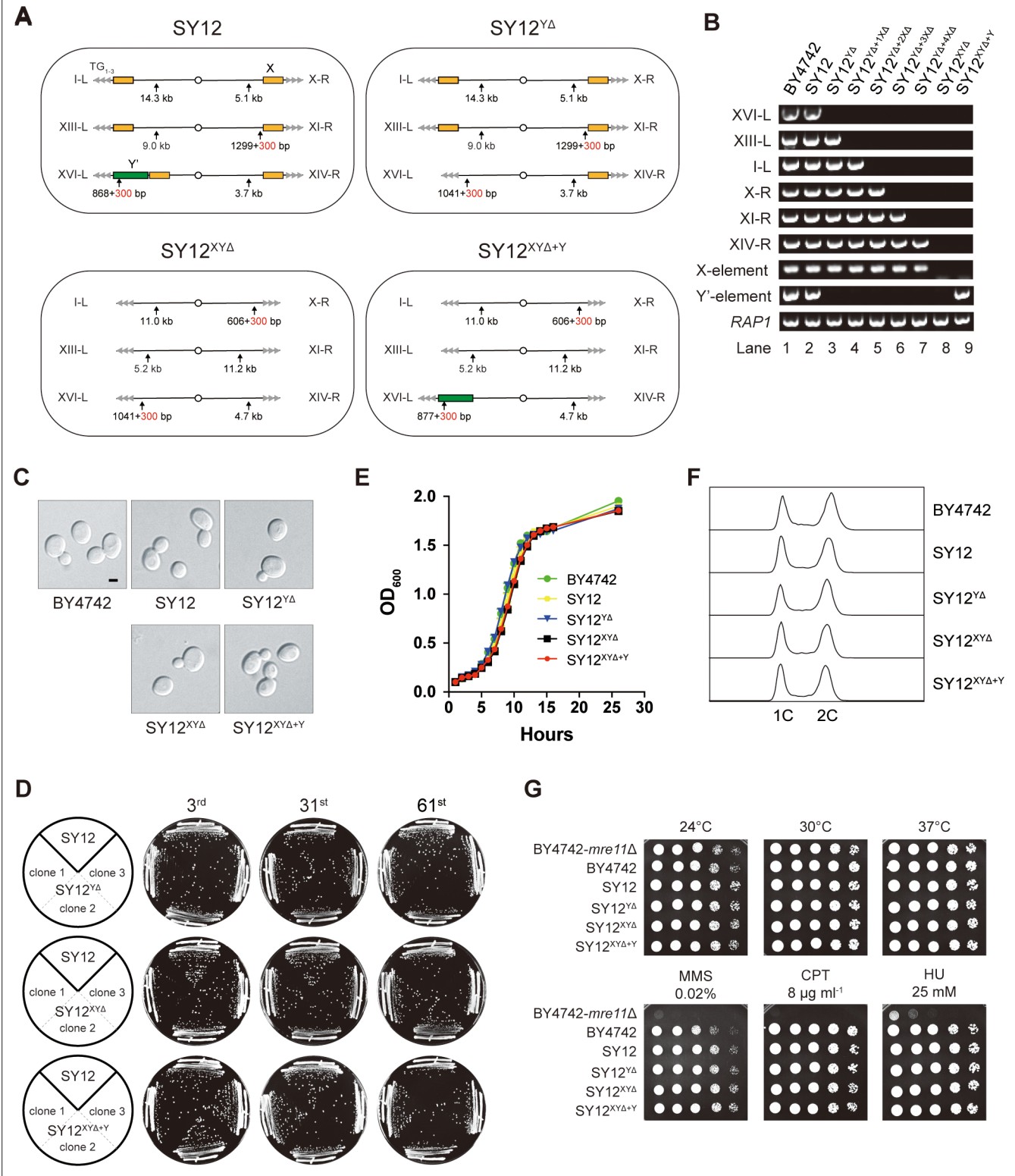

**Figure 3.** Characterization of SY12^YΔ, SY12^XYΔ, and SY12^XYΔ+T strains. (**A**) Schematic of chromosome structures in the SY12, SY12^YΔ, SY12^XYΔ, and SY12^XYΔ+T strains. Yellow box, X-element; green box, Y'-element; tandem gray triangles, telomeres. Vertical arrows and numbers indicate the positions and sizes of the sites and length of XhoI and PaeI-digested terminal fragments. (**B**) PCR analyses of the engineered sites of the individual telomeres (labeled on the left) in BY4742, SY12, SY12^YΔ, SY12^YΔ+1XΔ, SY12^YΔ+2XΔ, SY12^YΔ+3XΔ, SY12^YΔ+4XΔ, SY12^XYΔ, and SY12^XYΔ+T strains (labeled on top). Primer sequences for the PCR analyses are listed in ***Supplementary file 1***. *RAP1* was an internal control. (**C**) Morphology of BY4742, SY12, SY12^YΔ, SY12^XYΔ, and SY12^XYΔ+T cells in the

*Figure 3 continued on next page*

*Figure 3 continued*

exponential growth phase (30°C in YPD). Shown are DIC images. Scale bar, 2 μm. (**D**) Growth analysis of the SY12, SY12$^{Y\Delta}$, SY12$^{XY\Delta}$, and SY12$^{XY\Delta+T}$ strains. Several clones of the SY12, SY12$^{Y\Delta}$, SY12$^{XY\Delta}$, and SY12$^{XY\Delta+T}$ strains were re-streaked on YPD plates 61 times at intervals of 2 d. Shown were the 3rd, 31st, and 61st re-streaks. (**E**) Growth analysis of BY4742, SY12, SY12$^{Y\Delta}$, SY12$^{XY\Delta}$, and SY12$^{XY\Delta+T}$ cells in liquid culture. Error bars represent standard deviation (s.d.), n = 3. (**F**) Fluorescence-activated cell sorting (FACS) analysis of DNA content of BY4742, SY12, SY12$^{Y\Delta}$, SY12$^{XY\Delta}$, and SY12$^{XY\Delta+T}$ cells. (**G**) Dotting assays on YPD plates at low (24°C) and high (37°C) temperatures, or on YPD plates containing methyl methane sulfonate (MMS), camptothecin (CPT), or hydroxyurea (HU) at the indicated concentrations. The BY4742 *mre11Δ* haploid strain serves as a negative control because Mre11 is involved in the repair of double-stranded breaks (**Lewis et al., 2004**).

The online version of this article includes the following source data and figure supplement(s) for figure 3:

**Source data 1.** PCR identify of SY12 subtelomeric deletion strains in *Figure 3B*.

**Source data 2.** File containing *Figure 3B* and original scans of PCR identify of SY12 subtelomeric deletion strains.

**Source data 3.** Original file for the morphology analysis in *Figure 3C* for BY4742 strain.

**Source data 4.** Original file for the morphology analysis in *Figure 3C* for SY12 strain.

**Source data 5.** Original file for the morphology analysis in *Figure 3C* for SY12$^{Y\Delta}$ strain.

**Source data 6.** Original file for the morphology analysis in *Figure 3C* for SY12$^{XY\Delta}$ strain.

**Source data 7.** Original file for the morphology analysis in *Figure 3C* for SY12$^{XY\Delta+Y}$ strain.

**Source data 8.** File containing *Figure 3C* and original photos of morphology analysis of SY12 subtelomeric deletion strains.

**Source data 9.** Original file for the growth analysis in *Figure 3D* for SY12$^{Y\Delta}$ strain at the third streaks.

**Source data 10.** Original file for the growth analysis in *Figure 3D* for SY12$^{Y\Delta}$ strain at the 31st streaks.

**Source data 11.** Original file for the growth analysis in *Figure 3D* for SY12$^{Y\Delta}$ strain at the 61st streaks.

**Source data 12.** Original file for the growth analysis in *Figure 3D* for SY12$^{XY\Delta}$ strain at the third streaks.

**Source data 13.** Original file for the growth analysis in *Figure 3D* for SY12$^{XY\Delta}$ strain at the 31st streaks.

**Source data 14.** Original file for the growth analysis in *Figure 3D* for SY12$^{XY\Delta}$ strain at the 61st streaks.

**Source data 15.** Original file for the growth analysis in *Figure 3D* for SY12$^{XY\Delta+Y}$ strain at the third streaks.

**Source data 16.** Original file for the growth analysis in *Figure 3D* for SY12$^{XY\Delta+Y}$ strain at the 31st streaks.

**Source data 17.** Original file for the growth analysis in *Figure 3D* for SY12$^{XY\Delta+Y}$ strain at the 61st streaks.

**Source data 18.** File containing *Figure 3D* and original photos of growth analysis of SY12 subtelomeric deletion strains.

**Source data 19.** File containing output results of growth analysis of the SY12 subtelomeric deletion strains in *Figure 3E*.

**Source data 20.** Original FACS analysis results of *Figure 3F*.

**Source data 21.** Original file for the dotting assay on YPD plate at 24°C in *Figure 3G*.

**Source data 22.** Original file for the dotting assay on YPD plate at 30°C in *Figure 3G*.

**Source data 23.** Original file for the dotting assay on YPD plate at 37°C in *Figure 3G*.

**Source data 24.** Original file for the dotting assay on YPD plate containing MMS in *Figure 3G*.

**Source data 25.** Original file for the dotting assay on YPD plate containing CPT in *Figure 3G*.

**Source data 26.** Original file for the dotting assay on YPD plate containing HU in *Figure 3G*.

**Source data 27.** File containing *Figure 3G* and original photos of dotting assays of SY12 subtelomeric deletion strains.

**Figure supplement 1.** Schematics of CRISPR–Cas9-mediated deletion of X- and Y'-elements on individual chromosomes in SY12 cells.

expression upon *SIR2* deletion in SY12$^{Y\Delta}$, SY12$^{XY\Delta}$, and SY12$^{XY\Delta+Y}$ strains was more significant than that in the BY4742 or the SY12 strain, indicating that telomere silencing remains effective in the absence of X-and Y'-elements (*Figure 4B*). These findings align with previous studies showing that telomeres without an X- or Y'-element exert a position effect on the transcription of neighboring genes (*Aparicio et al., 1991*), and that X- and Y'-elements function as modulators of TPE (*Fourel et al., 1999*; *Lebrun et al., 2001*; *Ottaviani et al., 2008*).

In conclusion, the SY12$^{Y\Delta}$, SY12$^{XY\Delta}$, and SY12$^{XY\Delta+Y}$ strains behave similarly to the wild-type SY12 strain under all tested conditions (*Figures 3 and 4*). Their simplified telomere structure makes them potentially useful tools for telomere studies.

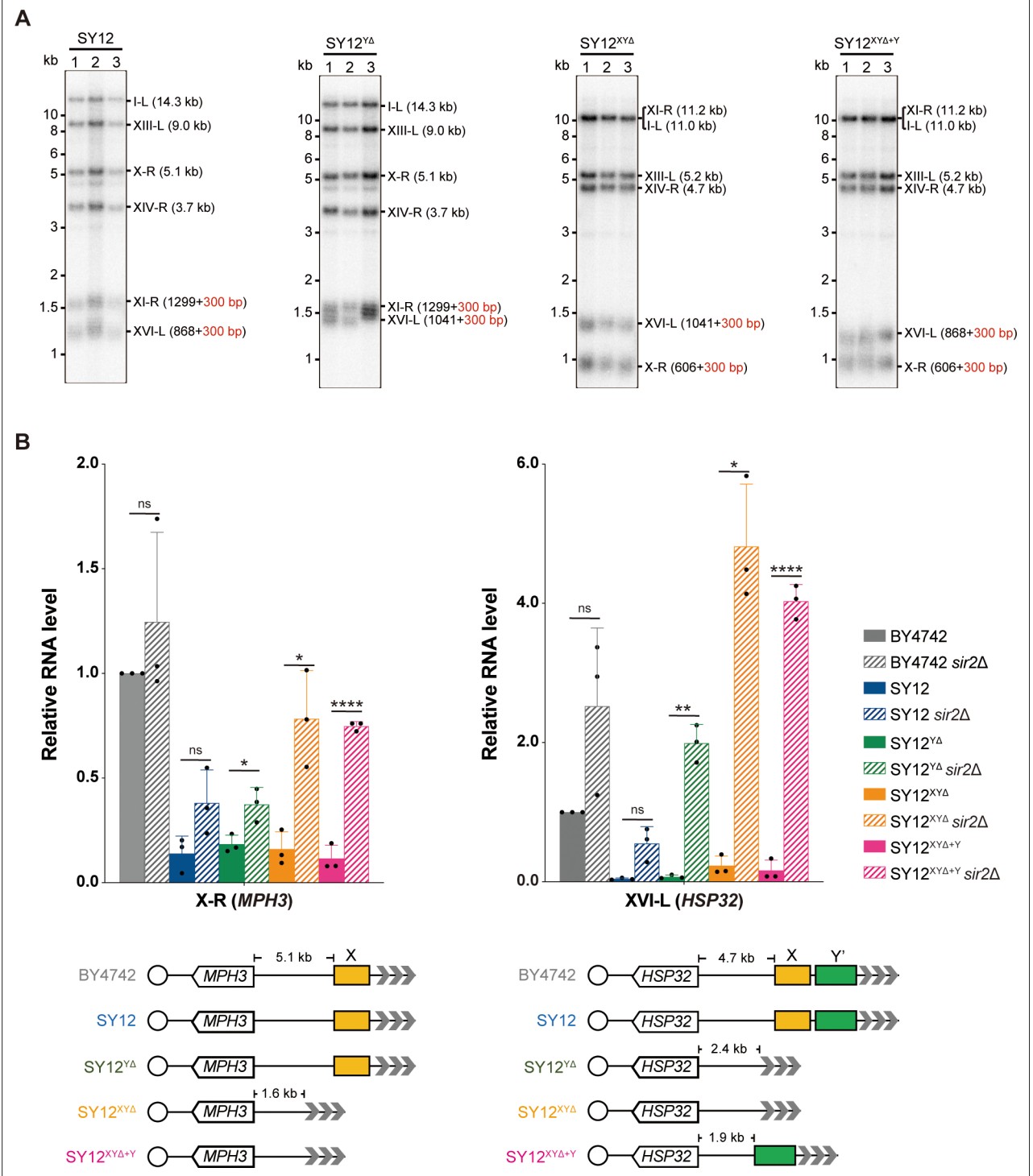

**Figure 4.** Telomere length and telomere silencing analyses of SY12<sup>YΔ</sup>, SY12<sup>XYΔ</sup>, and SY12<sup>XYΔ+Y</sup> strains. (**A**) Southern blotting analysis of telomere length in SY12, SY12<sup>YΔ</sup>, SY12<sup>XYΔ</sup>, and SY12<sup>XYΔ+Y</sup> (labeled on top) cells. Genomic DNA prepared from three independent clones of SY12, SY12<sup>YΔ</sup>, SY12<sup>XYΔ</sup>, and SY12<sup>XYΔ+Y</sup> strains were digested with XhoI and PaeI, and then subjected to Southern blotting with a TG$_{1-3}$ probe. The numbers in brackets indicate the telomere length of the corresponding chromosomes. (**B**) Expressions of *MPH3* and *HSP32* in BY4742, SY12, SY12<sup>YΔ</sup>, SY12<sup>XYΔ</sup>, and SY12<sup>XYΔ+Y</sup> cells were detected by qRT-PCR. The numbers above the schematic line (lower panels) indicate the distance to the corresponding subtelomeric elements or telomeres. The RNA levels of *MPH3* and *HSP32* were normalized by *ACT1*. The wild-type value is arbitrarily set to 1. Error bars represent standard deviation (s.d.), n = 3. 'ns', p>0.5 (Student's *t*-test); *p<0.05 (Student's *t*-test); **p<0.01 (Student's *t*-test); ****p<0.0001 (Student's *t*-test).

The online version of this article includes the following source data for figure 4:

*Figure 4 continued on next page*

*Figure 4 continued*

**Source data 1.** Original file for the Southern blotting analysis in *Figure 4A*.

**Source data 2.** File containing *Figure 4A* and original scans of the relevant Southern blotting analysis.

**Source data 3.** File containing output results of qPCR.

## Y'-elements are not strictly required for the formation of Type II survivors

The BY4742 strain harbors 19 Y'-elements distributed among 17 telomere loci. Numerous studies have emphasized the significance of Y'-elements in telomere recombination. For instance, Type I survivors exhibit significant amplification of Y'-elements (*Lundblad and Blackburn, 1993*; *Teng and Zakian, 1999*) and survivors show a marked induction of the potential DNA helicase Y'-Help1 encoded by Y'-elements (*Yamada et al., 1998*). Additionally, the acquisition of Y'-elements by short telomeres delays the onset of senescence (*Churikov et al., 2014*).

To investigate the requirement of Y'-elements in survivor formation, we deleted *TLC1* in SY12$^{Y\Delta}$ cells and conducted a cell viability assay. The results demonstrated that three individual colonies underwent senescence and subsequently recovered at different passages in liquid media (*Figure 5A*). Further analysis through Southern blotting revealed that the telomeres of SY12$^{Y\Delta}$ *tlc1Δ* cells underwent progressive shortening with each passage until reaching critically short lengths. Subsequently, TG$_{1-3}$ recombination occurred, leading to abrupt telomere elongation (*Figure 5B*).

Next, we examined the telomere patterns of 50 independent SY12$^{Y\Delta}$ *tlc1Δ* survivors using a multiple-colony streaking assay and Southern blotting analysis. Out of the 50 clones analyzed, no Type I survivors were detected due to the deletion of Y'-elements in SY12$^{Y\Delta}$ strain (*Figure 5C*). Two clones (labeled in orange, 4% of the survivors tested) displayed heterogeneous telomere tracts (*Figure 5C and D* and *Supplementary file 5*). Reintroduction of *TLC1* into a representative clone (named SY12$^{Y\Delta}$ *tlc1Δ*-T1) resulted in telomere length restoration similar to SY12$^{Y\Delta}$ cells (*Figure 5—figure supplement 1A*), indicating their classification as Type II survivors. Twenty-six clones (labeled in blue, 52% of the survivors tested) exhibited patterns identical to that of the SY12 *tlc1Δ* circular survivors (*Figures 5C and D* and *2D* and *Supplementary file 5*). Further mapping of erosion borders and sequencing of fusion junctions (*Figure 5E*, *Figure 5—figure supplement 2*, and *Supplementary file 3*) confirmed that three chromosomes from a randomly selected clone (named SY12$^{Y\Delta}$ *tlc1Δ*-C1) underwent intra-chromosomal fusions mediated by microhomology sequences. The erosion sites and fusion sequences differed from those observed in SY12 *tlc1Δ*-C1 cells (*Figure 2F*), suggesting the stochastic nature of intra-chromosome end fusion by MMEJ. As expected, the telomere Southern blotting pattern (XhoI digestion) of the SY12$^{Y\Delta}$ *tlc1Δ*-C1 survivor remained unchanged following telomerase reintroduction (*Figure 5—figure supplement 1B*). Further PFGE analysis confirmed that the chromosomes in SY12$^{Y\Delta}$ *tlc1Δ*-C1 were circulated (*Figure 2—figure supplement 5*). Notably, a significant proportion of the survivors displayed telomere signals that were different from those of either the Type II or circular survivors (labeled in black, 44% of the survivors tested, *Figure 5C and D* and *Supplementary file 5*), and they were uncharacterized survivors. Further deletion of *RAD52* in the SY12$^{Y\Delta}$ *tlc1Δ* cells affected, but did not eliminate, survivor generation (*Figure 5—figure supplement 3A*). Southern blotting assay confirmed that most of the recovered clones were circular survivors, and two were uncharacterized survivors (clones 9 and 16, labeled in black, *Figure 5—figure supplement 3B*), suggesting that survivor formation in SY12$^{Y\Delta}$ *tlc1Δ* rad52Δ cells does not strictly rely on the homologous recombination. Overall, these findings indicate that Y'-elements are not strictly required for Type II survivor formation (*Churikov et al., 2014*).

## X-elements are not strictly necessary for survivor generation

To investigate the contribution of X-elements to telomere recombination, we employed the SY12$^{XY\Delta+Y}$ strain, which contains only one Y'-element in the subtelomeric region, and the SY12$^{XY\Delta}$ *tlc1Δ* strain, which lacks both the X- and Y'-elements. Subsequently, we deleted *TLC1* in the SY12$^{XY\Delta+Y}$ and SY12$^{XY\Delta}$ strains and conducted a cell viability assay. Consistently, the deletion of *TLC1* in SY12$^{XY\Delta+Y}$ and SY12$^{XY\Delta}$ resulted in telomere shortening, senescence, and the formation of Type II survivors (*Figure 6—figure supplement 1*). Then, 50 independent clones of SY12$^{XY\Delta+Y}$ *tlc1Δ* or SY12$^{XY\Delta}$ *tlc1Δ* survivors were examined using Southern blotting (*Figure 6A and B*).

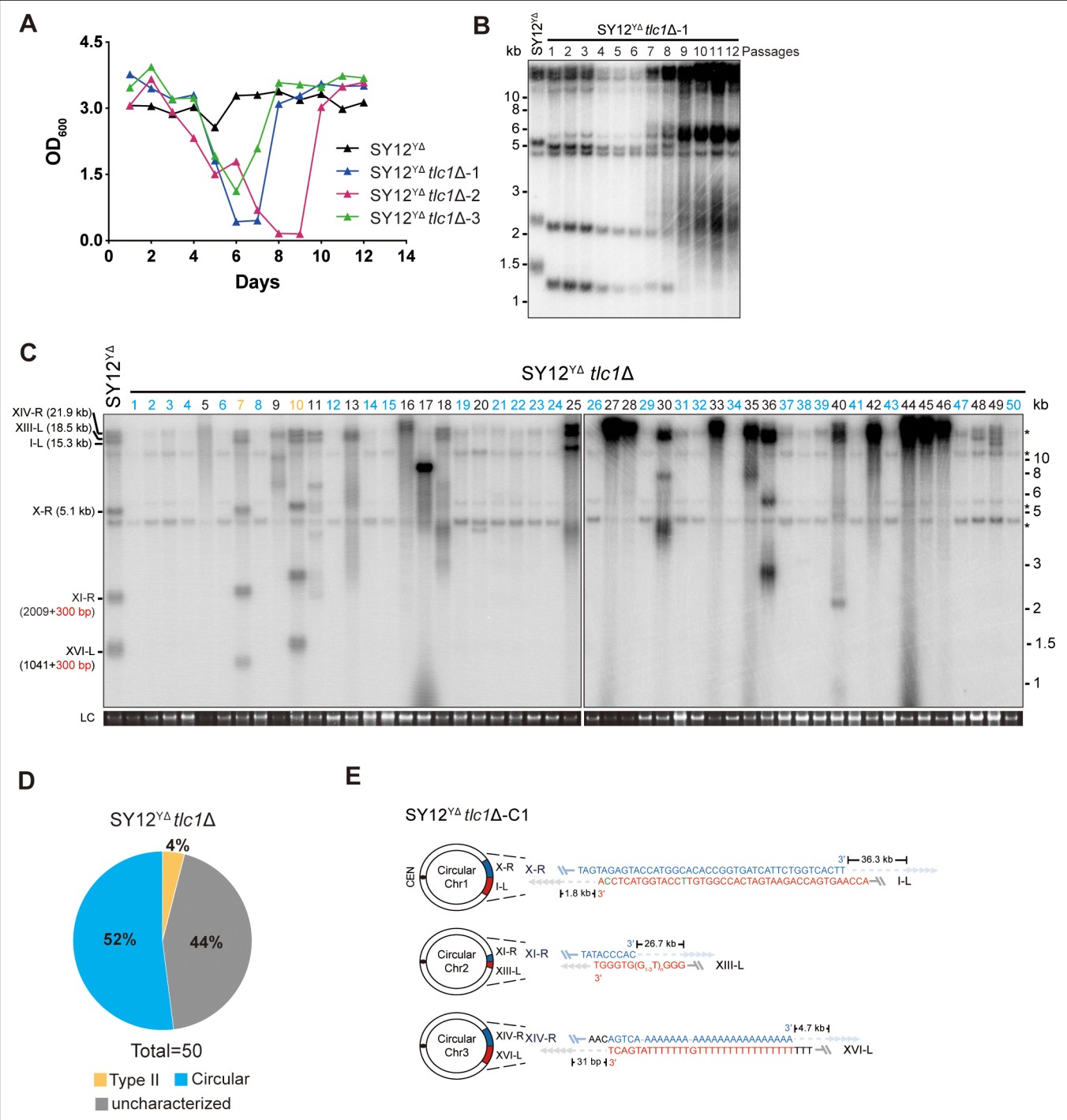

**Figure 5.** Survivor analysis of SY12^YΔ*tlc1*Δ strain. (**A**) Cell viability assay in liquid medium. The growth of SY12^YΔ (labeled in black) and SY12^YΔ*tlc1*Δ (three clones labeled in blue, purple, and green, respectively) strains were monitored every 24 hr for 12 d. (**B**) Telomeric Southern blotting assay of SY12^YΔ*tlc1*Δ survivors. Genomic DNAs prepared from SY12^YΔ*tlc1*Δ survivors assayed in (**A**) were digested with XhoI and subjected to Southern blotting with a TG$_{1-3}$ probe. (**C**) Telomere Southern blotting analysis of SY12^YΔ*tlc1*Δ survivors obtained on solid medium. Genomic DNAs of 50 independent survivors (labeled 1–50 on top) were digested with XhoI and hybridized by a TG$_{1-3}$ probe. Type II survivors: in orange; circular survivors: in blue; uncharacterized survivors: in black. Theoretical telomere restriction fragments of the SY12^YΔ strain are indicated on the left. LC: loading control. (**D**) The ratio of survivor types in SY12 ^YΔ*tlc1*Δ strain. n = 50; Type II, in orange; uncharacterized survivor, in gray; circular survivor, in blue. (**E**) Schematic of three circular chromosomes and fusion sequences in the SY12^YΔ*tlc1*Δ-C1 survivor. The sequence in blue indicates the sequences of X-R, XI-R, or XIV-R, the sequence in red indicates the sequences of I-L, XIII-L, or XVI-L. Bases in green are mis-paired, dashes are deleted. The numbers above or below the schematic line (chromosome) indicate the distance to the corresponding telomeres.

*Figure 5 continued on next page*

*Figure 5 continued*

The online version of this article includes the following source data and figure supplement(s) for figure 5:

**Source data 1.** File containing output results of growth analysis of the SY12$^{Y\Delta}$ *tlc1*Δ strain.

**Source data 2.** Original file for the Southern blotting analysis in *Figure 5B*.

**Source data 3.** File containing *Figure 5B* and original scans of the relevant Southern blotting analysis.

**Source data 4.** Original file for the Southern blotting analysis in *Figure 5C*.

**Source data 5.** Original file for the Southern blotting analysis in *Figure 5C*.

**Source data 6.** Original file for the loading control of Southern blotting analysis in *Figure 5C*.

**Source data 7.** Original file for the loading control of Southern blotting analysis in *Figure 5C*.

**Source data 8.** File containing *Figure 5C* and original scans of the relevant Southern blotting analysis.

**Source data 9.** File containing the original scans of the loading control of the Southern blotting analysis in *Figure 5C*.

**Figure supplement 1.** Southern blotting results of reintroducing *TLC1* into SY12 $^{Y\Delta}$ *tlc1*Δ survivors.

**Figure supplement 1—source data 1.** Original file for the Southern blotting analysis in *Figure 5—figure supplement 1A*.

**Figure supplement 1—source data 2.** Original file for the loading control of Southern blotting analysis in *Figure 5—figure supplement 1A*.

**Figure supplement 1—source data 3.** Original file for the Southern blotting analysis in *Figure 5—figure supplement 1B*.

**Figure supplement 1—source data 4.** Original file for the loading control of Southern blotting analysis in *Figure 5—figure supplement 1B*.

**Figure supplement 1—source data 5.** File containing *Figure 5—figure supplement 1A and B* and original scans of the relevant Southern blotting analysis.

**Figure supplement 1—source data 6.** File containing the original scans of the loading control of the Southern blotting analysis in *Figure 5—figure supplement 1*.

**Figure supplement 2.** PCR mapping of the borders of erosion in SY12$^{Y\Delta}$*tlc1*Δ-C1 cell.

**Figure supplement 3.** Survivor formation in SY12$^{Y\Delta}$ *tlc1*Δ *rad52*Δ strain.

**Figure supplement 3—source data 1.** File containing output results of growth analysis of the SY12$^{Y\Delta}$ *tlc1*Δ *rad52*Δ strain in *Figure 5—figure supplement 3A*.

**Figure supplement 3—source data 2.** Original file for the Southern blotting analysis in *Figure 5—figure supplement 3B*.

**Figure supplement 3—source data 3.** Original file for the loading control of Southern blotting analysis in *Figure 5—figure supplement 3B*.

**Figure supplement 3—source data 4.** File containing *Figure 5—figure supplement 3B* and original scans of the relevant Southern blotting analysis.

**Figure supplement 3—source data 5.** File containing the original scans of the loading control of the Southern blotting analysis in *Figure 5—figure supplement 3B*.

Among the SY12$^{XY\Delta+Y}$ survivors analyzed, 22 clones underwent chromosomal circularization (labeled in blue, 44% of the survivors tested, *Figure 6A and C* and *Supplementary file 5*). We randomly selected a clone named SY12$^{XY\Delta+Y}$ *tlc1*Δ-C1, and the results of erosion-border mapping and fusion junction sequencing showed that it had undergone intra-chromosomal fusions mediated by microhomology sequences (*Figure 6D*, *Figure 6—figure supplement 2*, and *Supplementary file 3*). Subsequently, Southern blotting revealed that the chromosome structure of SY12$^{XY\Delta+Y}$ *tlc1*Δ-C1 remained unchanged after *TLC1* reintroduction (*Figure 6—figure supplement 3*), and PFGE analysis confirmed the circular chromosome structure in SY12$^{XY\Delta+Y}$ *tlc1*Δ-C1 (*Figure 2—figure supplement 5*). Additionally, seven clones utilized the Type II recombination pathway and exhibited heterogeneous telomeric TG$_{1-3}$ tracts (labeled in orange, 14% of the survivors tested, *Figure 6A and C* and *Supplementary file 5*). Reintroduction of *TLC1* into a representative clone (named SY12$^{XY\Delta+Y}$ *tlc1*Δ-T1) restored the telomere length to normal (*Figure 6—figure supplement 3*). These findings indicate that the majority of cells underwent intra-chromosomal circularization or TG$_{1-3}$ recombination. While even though there is a Y'-element, no Type I survivors were generated in SY12$^{XY\Delta+Y}$ *tlc1*Δ survivors (*Figure 6A*). We speculated that the short TG$_{1-3}$ repeats located centromere-proximal to the Y'-elements play a crucial role in strand invasion and subsequent Y'-recombination. This speculation is consistent with a previous report stating that Type I events are virtually absent in the yeast strain Y55, which lacks TG$_{1-3}$ repeats centromere-proximal to the Y'-element (*Huang et al., 2001*). We also observed some clones displayed non-canonical telomere signals like SY12 *tlc1*Δ 'uncharacterized' survivors (labeled in black, 42% of the survivors tested, *Figure 6A and C* and *Supplementary file 5*). Overall, these data suggest that X-elements are not strictly necessary for survivor formation.

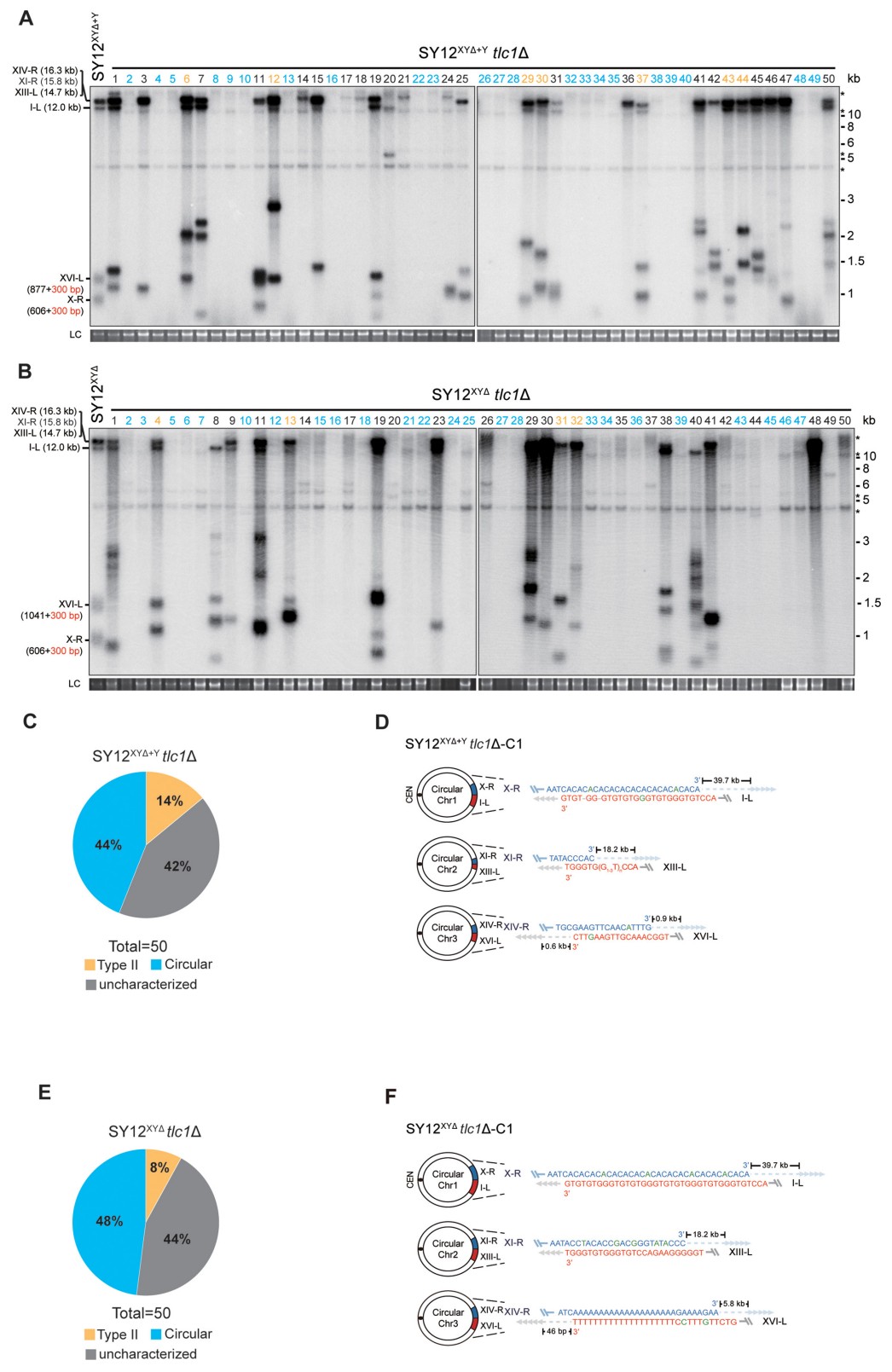

**Figure 6.** Survivor analysis of SY12$^{XYΔ}$tlc1Δ and SY12$^{XYΔ+Y}$tlc1Δ strains. (**A, B**) Telomere Southern blotting analysis of SY12$^{XYΔ+Y}$tlc1Δ (**A**) and SY12$^{XYΔ}$ tlc1Δ (**B**) survivors obtained on solid medium. 50 independent survivors (labeled 1–50 on top) were randomly picked, and their genomic DNAs were digested with XhoI and subjected to the Southern blotting assay with a TG$_{1-3}$ probe. Type II survivors: in orange; circular survivors: in blue; uncharacterized survivors:

*Figure 6 continued on next page*

*Figure 6 continued*

in black. The sizes of individual telomere restriction fragments of the SY12$^{XY\Delta+Y}$ and SY12$^{XY\Delta}$ strain are indicated on the left. LC: loading control. (**C, E**) The percentage of survivor types in SY12$^{XY\Delta+Y}$*tlc1*Δ (**C**) and SY12$^{XY\Delta}$*tlc1*Δ (**E**) strains. n = 50; Type II, in orange; uncharacterized survivor, in gray; circular survivor, in blue. (**D, F**) Schematic of three circular chromosomes and fusion sequences in the SY12$^{XY\Delta+Y}$*tlc1*Δ-C1 (**D**) and SY12$^{XY\Delta}$ *tlc1*Δ-C1 (**F**) survivors, respectively. The sequence in blue indicates the sequences of X-R, XI-R, or XIV-R, the sequence in red indicates the sequences of I-L, XIII-L, or XVI-L. Bases in green are mis-paired, dashes are deleted. The numbers above or below the schematic line (chromosome) indicate the distance to the corresponding telomeres.

The online version of this article includes the following source data and figure supplement(s) for figure 6:

**Source data 1.** Original file for the Southern blotting analysis in *Figure 6A*.

**Source data 2.** Original file for the Southern blotting analysis in *Figure 6A*.

**Source data 3.** Original file for the loading control of Southern blotting analysis in *Figure 6A*.

**Source data 4.** Original file for the loading control of Southern blotting analysis in *Figure 6A*.

**Source data 5.** File containing *Figure 6A* and original scans of the relevant Southern blotting analysis.

**Source data 6.** File containing the original scans of the loading control of the Southern blotting analysis in *Figure 6A*.

**Source data 7.** Original file for the Southern blotting analysis in *Figure 6B*.

**Source data 8.** Original file for the Southern blotting analysis in *Figure 6B*.

**Source data 9.** Original file for the loading control of Southern blotting analysis in *Figure 6B*.

**Source data 10.** Original file for the loading control of Southern blotting analysis in *Figure 6B*.

**Source data 11.** File containing *Figure 6B* and original scans of the relevant Southern blotting analysis.

**Source data 12.** File containing the original scans of the loading control of the Southern blotting analysis in *Figure 6B*.

**Figure supplement 1.** SY12$^{XY\Delta+Y}$*tlc1*Δ and SY12$^{XY\Delta}$*tlc1*Δ strains form Type II survivors in liquid culture.

**Figure supplement 1—source data 1.** File containing output results of growth analysis of the SY12$^{XY\Delta+Y}$ *tlc1*Δ strain in *Figure 6—figure supplement 1A*.

**Figure supplement 1—source data 2.** Original file for the Southern blotting analysis in *Figure 6—figure supplement 1B*.

**Figure supplement 1—source data 3.** File containing output results of growth analysis of the SY12$^{XY\Delta}$ *tlc1*Δ strain *Figure 6—figure supplement 1C*.

**Figure supplement 1—source data 4.** Original file for the Southern blotting analysis in *Figure 6—figure supplement 1D*.

**Figure supplement 1—source data 5.** File containing *Figure 6—figure supplement 1B and D* and original scans of the relevant Southern blotting analysis.

**Figure supplement 2.** PCR mapping of the borders of erosion in SY12$^{XY\Delta+T}$*tlc1*Δ-C1 cell.

**Figure supplement 3.** Southern blotting results of an SY12$^{XY\Delta+Y}$ *tlc1*Δ circular survivor and an SY12$^{XY\Delta+Y}$ *tlc1*Δ Type II survivor at the 20th re-streaks after *TLC1* reintroduction.

**Figure supplement 3—source data 1.** Original file for the Southern blotting analysis in *Figure 6—figure supplement 3*.

**Figure supplement 3—source data 2.** Original file for the loading control of Southern blotting analysis in *Figure 6—figure supplement 3*.

**Figure supplement 3—source data 3.** File containing *Figure 6—figure supplement 3* and original scans of the relevant Southern blotting analysis.

**Figure supplement 3—source data 4.** File containing the original scans of the loading control of the Southern blotting analysis in *Figure 6—figure supplement 3*.

**Figure supplement 4.** PCR mapping of the borders of erosion in SY12$^{XY\Delta}$*tlc1*Δ-C1 cell.

**Figure supplement 5.** Southern blotting results of reintroducing *TLC1* into SY12$^{XY\Delta}$*tlc1*Δ survivors.

**Figure supplement 5—source data 1.** Original file for the Southern blotting analysis in *Figure 6—figure supplement 5A*.

**Figure supplement 5—source data 2.** Original file for the loading control of Southern blotting analysis in *Figure 6—figure supplement 5A*.

*Figure 6 continued on next page*

*Figure 6 continued*

**Figure supplement 5—source data 3.** Original file for the Southern blotting analysis in *Figure 6—figure supplement 5B*.

**Figure supplement 5—source data 4.** Original file for the loading control of Southern blotting analysis in *Figure 6—figure supplement 5B*.

**Figure supplement 5—source data 5.** File containing *Figure 6—figure supplement 5A and B* and original scans of the relevant Southern blotting analysis.

**Figure supplement 5—source data 6.** File containing the original scans of the loading control of the Southern blotting analysis in *Figure 6—figure supplement 5A and B*.

**Figure supplement 6.** Survivor formation in SY12^XYΔ+Y *tlc1Δ rad52Δ* and SY12^XYΔ*tlc1Δ rad52Δ* strains.

**Figure supplement 6—source data 1.** File containing output results of growth analysis of the SY12^XYΔ+Y *tlc1Δ rad52Δ* strain in *Figure 6—figure supplement 6A*.

**Figure supplement 6—source data 2.** File containing output results of growth analysis of the SY12^XYΔ *tlc1Δ rad52Δ* strain in *Figure 6—figure supplement 6A*.

**Figure supplement 6—source data 3.** Original file for the Southern blotting analysis of the SY12^XYΔ+Y *tlc1Δ rad52Δ* strain in *Figure 6—figure supplement 6B*.

**Figure supplement 6—source data 4.** Original file for the Southern blotting analysis of the SY12^XYΔ *tlc1Δ rad52Δ* strain in *Figure 6—figure supplement 6B*.

**Figure supplement 6—source data 5.** Original file for the loading control of Southern blotting analysis of the SY12^XYΔ+Y *tlc1Δ rad52Δ* strain in *Figure 6—figure supplement 6B*.

**Figure supplement 6—source data 6.** Original file for the loading control of Southern blotting analysis the SY12^XYΔ *tlc1Δ rad52Δ* strain in *Figure 6—figure supplement 6B*.

**Figure supplement 6—source data 7.** File containing *Figure 6—figure supplement 6B* and original scans of the relevant Southern blotting analysis.

**Figure supplement 6—source data 8.** File containing the original scans of the loading control of the Southern blotting analysis in *Figure 6—figure supplement 6B*.

**Figure supplement 7.** Survivor formation in SY12^XYΔ*tlc1Δ rad51Δ* strain.

**Figure supplement 7—source data 1.** Original file for the Southern blotting analysis in *Figure 6—figure supplement 7A*.

**Figure supplement 7—source data 2.** File containing *Figure 6—figure supplement 7A* and original scans of the relevant Southern blotting analysis.

Among the SY12^XYΔ survivors, 24 displayed a 'circular survivor' pattern (labeled in blue, 48% of the survivors tested, *Figure 6B and E* and *Supplementary file 5*). Additional PCR-sequencing assays and PFGE analysis of the SY12^XYΔ *tlc1Δ*-C1 cells confirmed the occurrence of intra-chromosomal fusions mediated by microhomology sequences (*Figure 6F*, *Figure 6—figure supplement 4*, *Supplementary file 3*, and *Figure 2—figure supplement 4*). Reintroduction of *TLC1* into a representative clone named SY12^XYΔ*tlc1Δ*-C1 could restore its telomere length to WT level (*Figure 6—figure supplement 5A*). Also, 4 of 50 survivors harbored Type II telomere structure (labeled in orange, 8% of the survivors tested, *Figure 6B and E* and *Supplementary file 5*). Reintroduction of *TLC1* into a representative clone named SY12^XYΔ*tlc1Δ*-T1 could restore its telomere length to WT level (*Figure 6—figure supplement 5B*). Some of the survivors (labeled in black, 44% of the survivors tested, *Figure 6B and E* and *Supplementary file 5*) were not characterized. Like in SY12 *tlc1Δ* cells, Rad52 is not strictly required for the formation of circular survivors in SY12^XYΔ *tlc1Δ rad52Δ* and SY12^XYΔ+Y *tlc1Δ rad52Δ* strains (*Figure 6—figure supplement 6A and B*). To investigate whether Type I-specific mechanisms are still utilized in the survivor formation in Y'-less strain, we deleted *RAD51* in SY12^XYΔ *tlc1Δ*, and found that SY12^XYΔ *tlc1Δ rad51Δ* strain was able to generate three types of survivors, including Type II survivor, circular survivor, and uncharacterized survivor (*Figure 6—figure supplement 7A*), similar to the observations in SY12^XYΔ *tlc1Δ* strain (*Figure 6B*). Notably, the proportions of circular and uncharacterized survivors in the SY12^XYΔ *tlc1Δ rad51Δ* strain were 36% (9/25) and 32% (8/25) (*Figure 6—figure supplement 7B* and *Supplementary file 5*), respectively, lower than 48% and 44% in the SY12^XYΔ *tlc1Δ* strain (*Figure 6E* and *Supplementary file 5*). Accordingly, the ratio of Type II survivor in SY12^XYΔ *tlc1Δ rad51Δ* was (32% of the survivors tested, *Figure 6—figure supplement 7B* and *Supplementary file 5*)

was higher than SY12$^{XY\Delta}$ *tlc1Δ* strain (8% of the survivors tested, *Figure 6E* and *Supplementary file 5*), suggesting that Type I-specific mechanisms still contribute to the survivor formation even in the Y'-less strain SY12$^{XY\Delta}$. Collectively, the aforementioned data suggest that X-elements, as well as Y'-elements, are not essential for the generation of Type II survivors.

## Discussion

The wild-type yeast strain BY4742, commonly used in laboratories, possesses 19 Y'-elements at 17 telomere loci and 32 X-elements at 32 telomere loci. This abundance of Y'-elements and X-elements poses challenges for loss-of-function studies, highlighting the need for a strain lacking all Y'-elements and X-elements. Fortunately, we have previously constructed the single-chromosome yeast strain SY14, which contains only one copy of Y'-element and two copies of X-element (*Shao et al., 2018*), and could have been an ideal tool. However, the telomerase-null survivors of SY14 mainly bypassed senescence through chromosomal circularization, providing limited insights into the roles of Y'- and X-elements in telomere maintenance (*Wu et al., 2020*). Therefore, in this study, we employed the SY12 strain, which has three chromosomes, to investigate the functions of Y'- and X-elements at telomeres (*Figure 2A*, left panel).

We constructed the SY12$^{Y\Delta}$, SY12$^{XY\Delta+Y}$, and SY12$^{XY\Delta}$ strains, which lack the Y'-element, X-elements, and both X- and Y'-elements, respectively (*Figure 3A*). Surprisingly, the SY12$^{Y\Delta}$, SY12$^{XY\Delta}$, and SY12$^{XY\Delta+Y}$ strains exhibited minimal defects in cell proliferation, genotoxic sensitivity, and telomere homeostasis (*Figures 3 and 4*). These results demonstrate, for the first time, that both X- and Y'-elements are dispensable for cellular functions. Thus, the SY12$^{Y\Delta}$, SY12$^{XY\Delta}$, and SY12$^{XY\Delta+Y}$ strains established in this study, with their simplified telomere structures, are valuable resources for telomere biology research.

Subtelomeric regions are known to be highly variable and often contain species-specific homologous DNA sequences. In the case of fission yeast, subtelomeric regions consist of subtelomeric homologous (SH) and telomere-distal sequences. Previous studies have shown that SH sequences in fission yeast do not significantly impact telomere length, mitotic cell growth, or stress responses. However, they do play a role in buffering against the spreading of silencing signals from the telomere (*Tashiro et al., 2017*). Though the 'core X' sequence acts as a protosilencer (*Lebrun et al., 2001*), the X-STRs and Y'-STAR possess anti-silencing properties that limit the spreading of heterochromatin in budding yeast (*Fourel et al., 1999*), the telomere position effect remains effective in the strains that lack both X- and Y'-elements (*Figure 4B*). Given the remarkable differences in both sequence and size between the subtelomeric regions of budding yeast and fission yeast, it is difficult to compare the extent to which subtelomeric elements affect telomere silencing.

Amplification of Y'-element(s) is a characteristic feature of canonical Type I survivors. Type I survivors emerged in SY12 strain, indicating that multiple Y'-elements in tandem are not strictly required for Type I recombination (*Figure 2D*). Interestingly, the telomerase-null SY12$^{Y\Delta}$ and SY12$^{XY\Delta}$ cells, lacking Y'-elements, failed to generate Type I survivors but could generate Type II survivors, indicating that the acquisition of Y'-elements is not a prerequisite for Type II survivor formation (*Figures 5C* and *6B*). These observations support the notion that Type I and Type II survivors form independently, although both may utilize a common alternative telomere-lengthening pathway (*Kockler et al., 2021*). Moreover, a subset of SY12 *tlc1Δ*, SY12$^{Y\Delta}$ *tlc1Δ*, SY12$^{XY\Delta+Y}$ *tlc1Δ*, and SY12$^{XY\Delta}$ *tlc1Δ* cells could escape senescence and become survivors through microhomology-mediated intra-chromosomal end-to-end fusion (chromosome circularization) (*Figures 2D*, *5C*, and *6A and B*, labeled in blue). Notably, the survivors with all circular chromosomes were readily recovered from the telomerase-null SY11 to SY14, but not SY1 to SY10 cells (*Figure 1*). Several reasons could account for this. First, a smaller number of telomeres provides fewer recombination donors and acceptors, resulting in less efficient inter-chromosomal homologous recombination (e.g., TG$_{1-3}$ tracts recombination or Y'-element acquisition). Second, the continuously shortened telomeres of linear chromosomes may trigger another round of senescence, while survivors with circular chromosomes do not encounter end-replication problems and therefore exhibit greater stability. Third, the presence of homologous sequences at both chromosome ends appears to be a minimum requirement for microhomology-mediated intra-chromosomal end-to-end fusion. With fewer homologous sequences, the probability of chromosome circularization decreases, and with more chromosomes, the likelihood of circularizing each chromosome within a cell diminishes. Fourth, in cells with fewer telomeres, intra-chromosomal telomere fusions are more likely to occur, while lethal inter-chromosomal fusions are competed out. However, we can speculate that in

telomerase-null cells with eroded chromosome ends, stochastic repair mechanisms such as homologous recombination, microhomology-mediated end joining, and inter- and intra-chromosomal fusions operate simultaneously. Only those survivors that maintain a relatively stable genome and robust growth can be experimentally recovered.

*S. cerevisiae* (budding yeast) and *Schizosaccharomyces pombe* (fission yeast) are the most commonly used laboratory systems, separated by approximately 1 Gya (billion years ago) according to molecular-clock analyses (*Hedges, 2002*). Despite both species having genomes are both over 12 megabases in length, haploid *S. cerevisiae* contains 16 chromosomes, while *S. pombe* has only 3 chromosomes (*Forsburg, 2005*). The telomerase-independent mechanisms for maintaining chromosome ends differ between these two yeasts. In budding yeast, homologous recombination is the primary mode of survival in telomerase-deficient cells, resulting in the generation of Type I or Type II survivors (*McEachern and Haber, 2006*). Telomerase- and recombination-deficient cells occasionally escape senescence through the formation of palindromes at chromosome ends in the absence of *EXO1* (*Maringele and Lydall, 2004*). Fission yeast cells lacking telomerase can also maintain their chromosome termini by recombining persistent telomere sequences, and survivors with all intra-circular chromosomes (*Nakamura et al., 1998*) or intermolecular fusions (*Tashiro et al., 2017*; *Wang and Baumann, 2008*) have been observed. In our research, some SY12 *tlc1Δ* cells, which have three chromosomes, also bypassed senescence by circularizing their chromosomes (*Figure 2D*), suggesting that a lower chromosome number increases the likelihood of recovering survivors containing circular chromosomes.

While most eukaryotes employ telomerase for telomere replication, some eukaryotes lack telomerase and utilize recombination as an alternative means to maintain telomeres (*Biessmann and Mason, 1997*). In *Drosophila*, telomeres are replicated through a retrotransposon mechanism (*Levis et al., 1993*; *Louis, 2002*). The structure and distribution of Y'-elements in *S. cerevisiae* suggest their origin from a mobile element (*Jäger and Philippsen, 1989*; *Louis and Haber, 1992*), and Y'-elements can be mobilized through a transposition-like RNA-mediated process (*Maxwell et al., 2004*). In telomerase-deficient yeast cells, homologous recombination can acts as a backup mechanism for telomere replication (*Lundblad and Blackburn, 1993*), and the reintroduction of telomerase efficiently inhibits telomere recombination and dominates telomere replication (*Chen et al., 2009*; *Peng et al., 2015*; *Teng and Zakian, 1999*), These findings suggest that subtelomeric region amplification mediated by recombination and/or transposition may represent ancient telomere maintenance mechanisms predating the evolution of telomerase (*de Lange, 2004*). Therefore, subtelomeric X- and Y'-elements might be considered as evolutionary 'fossils' in the *S. cerevisiae* genome, and their elimination has little impact on telomere essential functions and genome stability.

## Materials and methods

### Yeast strains and plasmids

Yeast strains used in this study are listed in *Supplementary file 6*. The plasmids for gene deletion and endogenous expression of *TLC1* were constructed based on the pRS series as described previously (*Sikorski and Hieter, 1989*). We use PCR to amplify the upstream and downstream sequence adjacent to the target gene, and then the PCR fragments were digested with different restriction enzymes and inserted into pRS plasmids. Plasmids were introduced into budding yeast by standard procedures, and transformants were selected on auxotrophic medium (*Orr-Weaver et al., 1981*).

### Multiple-colony streaking assay

Single clones of indicated yeast strains were randomly picked and streaked on extract-peptone-dextrose (YPD) plates. Thereafter, several clones of their descendants were passaged by successive re-streaks at 30°C. This procedure was repeated dozens of times every 2 d.

### Telomere Southern blotting

Southern blotting was performed as previously described (*Hu et al., 2013*). Yeast genomic DNA was extracted by a phenol chloroform method. Restriction fragments were separated by electrophoresis in 1% agarose gel, transferred to Amersham Hybond-N$^+$ membrane (GE Healthcare), and hybridized with α-$^{32}$P dCTP-labeled probe.

## Cell viability assay

Cell viability assay was performed as previously described with a few modifications (*Le et al., 1999*). Three independent single colonies of indicated strains were grown to saturation at 30°C. Then the cell density was measured every 24 hr by spectrometry ($OD_{600}$), and the cultures were diluted to the density at $OD_{600} = 0.01$. This procedure was repeated several times to allow the appearance of survivors. The genomic DNA samples at indicated time points were harvested for telomere length analysis.

## Molecular analysis of circular chromosomes

Fusion events were determined by PCR amplification and DNA sequencing. Genomic DNA was extracted by phenol chloroform. First, we use primers pairs located at different sites of each chromosome arm at an interval of 1 kb (listed in *Supplementary file 1*) to determine the erosion site of each chromosome; PCR was performed as standard procedures in 10 µl reactions by TaKaRa Ex Taq. To amplify the sequence of fusion junction, we use pairs of primers oriented to different arm of each chromosome; PCR was performed as standard procedures in 50 µl reactions by TaKaRa LA Taq. The fragments were purified by kit (QIAGEN), then they were sequenced directly or cloned into the pMD18-T Vector (TaKaRa) for sequencing.

## CRISPR-Cas9-mediated X- and Y'-elements deletion

X- and Y'-elements were deleted as described (*Shao et al., 2018*; *Shao et al., 2019*). Briefly, pgRNA and a DNA targeting cassette, containing a selection marker, a homology arm (DR1), a direct repeat (DR2), and telomeric repeats, were co-introduced into indicated cells harboring pCas9. pCas9 nuclease was directed to a specific DNA sequence centromere-proximal to the subtelomeric region with the guidance of gRNA1, where it induces a double-stranded break. Homologous recombination between the broken chromosome and the provided DNA targeting cassette caused the deletion of X- and Y'-elements. The positive transformants identified by PCR were transferred into the galactose-containing liquid medium, which induces the expression of the gRNA2 on pCas9 to cut at the target site near the *URA3* gene and on the backbone of pgRNA. Then the culture was plated on the medium containing 5'-FOA to select for eviction of the *URA3* marker.

## Cell growth assay

Three individual colonies of the indicated strains were inoculated into 5 ml liquid medium and incubated at 30°C. The cell cultures were then diluted in 30 ml of fresh YPD medium to the density at $OD_{600} = 0.1$. Then the density of cells was measured by spectrometry ($OD_{600}$) hourly.

## FACS assay

The FACS analysis was performed as previously described (*He et al., 2019*). Yeast cells were cultured at 30°C until the log phase, and then 1 ml of the cells was harvested. The cells were washed with cold sterile ddH$_2$O and fixed with 70% ethanol overnight at 4°C. The following day, the cells were washed with 50 mM sodium citrate buffer (pH 7.2) and then digested with 0.25 mg/ml RNase A at 37°C for 2–3 hr, followed by 0.2 mg/ml Protease K at 50°C for 1 hr. Both RNase A and Protease K were diluted in sodium citrate buffer. The cells were resuspended in 500 µl sodium citrate buffer and then sonicated for 45 s at 100% power. The DNA of the cells was stained with 20 µg/ml propidium iodide (PI) at 4°C overnight or at room temperature for 1 hr. FACS analysis was performed on a BD LSRII instrument.

## Serial dilution assay

A single colony per strain was inoculated into 3 ml liquid medium and incubated at 30°C. The cell cultures were then adjusted to a concentration of $OD_{600} \sim 0.5$. Fivefold serially diluted cells were spotted on the indicated plates. The plates were incubated at 30°C for the appropriate time prior to photography.

## RNA extraction and RT-qPCR

Three independent single colonies of indicated strains were grown to log phase at 30°C. Yeast pellets from a 1 ml cell culture were digested with Zymolyase 20T (MP Biomedicals, LLC) to obtain spheroplasts. RNA was extracted with RNeasy mini kit (QIAGEN) followed by reverse transcription using the Fastquant RT kit (Tiangen). Real-time PCR was carried out using SYBR Premix Ex Taq II (Takara) on

the Applied Biosystems StepOne Real-Time PCR System. Primer pairs used in RT-qPCR are listed in *Supplementary file 1*. The gene expression levels were normalized to that of *ACT1* and the wild-type value is arbitrarily set to 1.

## PFGE analysis

DNA plugs for PFGE were prepared according to the manufacturer's instructions (Bio-Rad) and *Ishii et al., 2008*. Fresh yeast cells were inoculated in 50 ml YPD and incubated at 30°C until the $OD_{600}$ reached approximately 1.0. The cells were subsequently harvested, washed twice with cold EDTA buffer (50 mM, pH 8.0), and resuspended in 300 µl of CSB buffer (10 mM pH 7.2 Tris–Cl, 20 mM NaCl, 100 mM pH 8.0 EDTA, 4 mg/ml lyticase) and blended with 300 µl of 2% low-melt agarose (Bio-Rad). Then, 100 µl of resuspended cells were added to each plug and incubated at 4°C for 30 min until the agarose plugs were solidified. The solidified agarose plugs were incubated in lyticase buffer (10 mM pH 7.2 Tris–Cl, 100 mM pH 8.0 EDTA, 1 mg/ml lyticase) at 37°C for 3 hr, followed by incubation in Proteinase K Reaction Buffer (100 mM pH 8.0 EDTA, 0.2% sodium deoxycholate, 1% sodium lauryl sarcosine) containing 1 mg/ml Proteinase K at 50°C for 12 hr. The plugs were washed four times in 25 ml of wash buffer (20 mM Tris, pH 8.0, 50 mM EDTA) for 1 hr each time at room temperature with gentle agitation. The plugs were then fixed into a pulsed field agarose gel (Bio-Rad), and the CHEF-DR II Pulsed Field Electrophoresis System (Bio-Rad) was used for gel electrophoresis. The electrophoresis conditions for separation were as follows: 0.8% agarose gel, 1× TBE buffer, 14°C temperature, first run: initial switch time 1200 s; final switch time 1200 s; run time 24 hr; voltage gradient 2 V/cm; angle 96°; second run: initial switch time 1500 s; final switch time 1500 s; run time 24 hr; voltage gradient 2 V/cm; angle 100°; third run: initial switch time 1800 s; final switch time 1800 s; run time 24 hr; voltage gradient 2 V/cm; angle 106°. The gel was stained with GelstainRed nucleic acid dye (US Everbright), and PFGE Gels were imaged by Tanon 2500.

## Acknowledgements

We thank the members of Zhou lab for discussions and suggestions for this project.

## Additional information

### Funding

| Funder | Grant reference number | Author |
| --- | --- | --- |
| National Key Research and Development Program of China | 2023YFA0913400 | Jin-Qiu Zhou |
| The National Natural Science Foundation of China | 32150004 | Jin-Qiu Zhou |

The funders had no role in study design, data collection and interpretation, or the decision to submit the work for publication.

### Author contributions

Can Hu, Xue-Ting Zhu, Investigation, Writing – review and editing; Ming-Hong He, Writing – review and editing; Yangyang Shao, Zhongjun Qin, Methodology; Zhi-Jing Wu, Conceptualization, Investigation, Methodology, Writing – original draft, Writing – review and editing; Jin-Qiu Zhou, Conceptualization, Supervision, Writing – original draft, Writing – review and editing

### Author ORCIDs

Jin-Qiu Zhou https://orcid.org/0000-0003-1986-8611

Reviewer #1 (Public Review): https://doi.org/10.7554/eLife.91223.4.sa1
Reviewer #2 (Public Review): https://doi.org/10.7554/eLife.91223.4.sa2
Reviewer #3 (Public Review): https://doi.org/10.7554/eLife.91223.4.sa3

Author response https://doi.org/10.7554/eLife.91223.4.sa4

## Additional files

### Supplementary files
- Supplementary file 1. Primer used in this study.
- Supplementary file 2. The remaining subtelomeric elements in SY8 to SY13 strains.
- Supplementary file 3. Details of fusion points in the circular survivors.
- Supplementary file 4. Details of the creation of SY12 subtelomeric engineered strains.
- Supplementary file 5. Quantitation of each survivor type in SY12 subtelomeric engineered strains.
- Supplementary file 6. Yeast strains used in this study.
- MDAR checklist

### Data availability
All data generated or analysed during this study are included in the manuscript and supporting files; source data files have been provided for all figures.

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
