## [Editor Report · eLife assessment]

This **important** study advances our understanding of the biological significance of the DNA sequence adjacent to telomeres. The data presented **convincingly** demonstrate that subtelomeric repeats are non-essential and have a minimal, if any, role in maintaining telomere integrity of budding yeast. The work will be of interest to the telomere community specifically and the genome integrity community more broadly.

---

## [Referee Report · Reviewer #1 (Public Review)]

The authors have generated a set of yeast *S. cerevisiae* strains containing different numbers of chromosomes.

Elimination of telomerase activates homologous recombination (HR) to maintain telomeres in cells containing the original 16 chromosomes. However, elimination of telomerase leads to circularization of cells containing a single or two chromosomes. The authors examined whether the subtelomeric sequences X and Y' promote HR-mediated telomere maintenance using the strain SY12 carrying three chromosomes. They found that the subtelomeric sequences X and Y' are dispensable for cell proliferation and HR-mediated telomere maintenance in telomerase-minus SY12 cells. They conclude that subtelomeric X and Y' sequences do not play essential roles in both telomerase-proficient and telomerase-null cells and propose that these sequences represent remnants of genome evolution.

Interestingly, telomerase-minus SY12 generates survivors that are different from well-established Type I or Type II survivors. The authors uncover atypical telomere formation which does not depend on the Rad52 homologous recombination pathway.

Strengths:

The authors examined whether the subtelomeric sequences X and Y' promote HR-mediated telomere maintenance using the strain SY12 carrying three chromosomes. They show that subtelomeres do not have essential roles in telomere maintenance and cell proliferation.

Weaknesses:

It is not fully addressed how atypical survivors are generated independently of Rad52-mediated homologous recombination.

It remains possible that X and Y elements influence homologous recombination, type 1 and type 2 (type X), at telomeres. In particular, the presence of X and Y elements appears to be important for promoting type 1 recombination, although the authors conclude "Elimination of subtelomeric repeat sequences exerts little effect on telomere functions".

---

## [Referee Report · Reviewer #2 (Public Review)]

Summary:

In this work, Hu and colleagues investigate telomerase-independent survival in *Saccharomyces cerevisiae* strains engineered to have different chromosome numbers. The authors describe the molecular patterns of survival that change with fewer chromosomes and that differ from the well-described canonical Type I and Type II, including chromosome circularization and other atypical outcomes. They then take advantage of the strain with 3 chromosomes to examine the effect of deleting all the subtelomeric elements, called X and Y'. For most of the tested phenotypes, they find no significant effect of the absence of X- and Y'-element, and show that they are not essential for survivor formation. They speculate that X- and Y'-elements are remnants of ancient telomere maintenance mechanisms.

Strengths:

This work advances our understanding of the telomerase-independent strategies available to the cell by altering the structure of the genome and of the subtelomeres, a feat that was enabled by the set of strains they engineered previously. By using strains with non-standard genome structures, several alternative survival mechanisms are uncovered, revealing the diversity and plasticity of telomere maintenance mechanisms. Overall, the conclusions are well supported by the data, with adequate sample sizes for investigating survivors. The assessment of the genetic requirements for survivors in strains with different chromosome numbers greatly improved the quality of this work. The molecular analyses based on Southern blots are also very well-conducted.

Weaknesses:

The authors discovered alternative telomerase-independent survival strategies beyond the well-described type I and II (including circularization, type X and atypical, as they called them) at play in the context of reduced number of chromosomes. Their work provides a molecular and a partial genetic characterization of these survival pathways. A more thorough analysis of the frequency of each type of survivors and their genetic requirements would have advanced our understanding or the diversity of survival strategies in the absence of telomerase. However, as noted by the authors, the quantification of the rate of emergence of survivors (and their subtypes) is very difficult to achieve. This comment is therefore not meant as a criticism but rather as a perspective on exciting future research avenues.

---

## [Referee Report · Reviewer #3 (Public Review)]

This study investigates subtelomeric repetitive sequences in the budding yeast *Saccharomyces cerevisiae*, known as Y' and X-elements. Taking advantage of yeast strain SY12 that contains only 3 chromosomes and six telomeres (normal yeast strains contain 32 telomeres) the authors are able to generate a strain completely devoid of Y'- and X-elements.

Strengths:

They demonstrate that the SY12 delta XY strain displays normal growth, with stable telomeres of normal length that were transcriptionally silenced, a key finding with wide implications for telomere biology. Inactivation of telomerase in the SY12 and SY12 delta XY strains frequently resulted in survivors that had circularized all three chromosomes, hence bypassing the need for telomeres altogether. They show that survivors with fused chromosomes and so-called atypical survivors arise independently of the central recombination protein Rad52. The SY12 and SY12 delta XY yeast strains can become a useful tool for future studies of telomere biology. The conclusions of this manuscript are well supported by the data and are valuable for researchers studying telomeres.

Weaknesses:

A weakness of the manuscript is the analysis of telomere transcriptional silencing. They state: "The results demonstrated a significant increase in the expression of the MPH3 and HSP32 upon Sir2 deletion, indicating that telomere silencing remains effective in the absence of X and Y'-elements". However, for the SY12 strain, their analyses indicate that the difference between the WT and sir2 strains is nonsignificant. In addition, a striking observation is that the SY12 strain (with only three chromosomes) express much less of both MPH3 and HSP32 than the parental strain BY4742 (16 chromosomes), both in the presence and absence of Sir2.

---

## [Author Response]

The following is the authors’ response to the previous reviews.

**Reviewer #1**
The authors provided experimental data in response to my comments/suggestions in the revision. Overall, most points were appropriate and satisfactory, but some issues remain.(1) It is not fully addressed how atypical survivors are generated independently of Rad52-mediated homologous recombination.The newly provided data indicate that the formation of atypical telomeres is independent of the Rad52 homologous recombination pathway."The atypical telomeres clones exhibit non-uniform telomere pattern", but the TG-hybridized signals after XhoI digestion are clear and uniform."Atypical telomere" clones may carry circular chromosomes embedded with short TG repeats, rather than linear chromosomes. In other words, atypical telomeres may differ from telomeres, the ends of chromosomes. Is atypical telomere formation dependent on NHEJ? Given that "two chromosomes underwent intra-chromosomal fusions" (Line 248), are atypical telomere clones detected frequently in SY13 cells containing two chromosomes?

We thank the reviewer’s questions. Frankly, we have not been able to determine the chromosome structures in these so-called "atypical survivors". As we mentioned in the manuscript, there could be mixed telomere structures, e.g. TG tract amplification, intro-chromosome telomere fusion and inter-chromosome telomere fusion. Worse still, these 'atypical survivors' may not have maintained a stable genome, and their karyotype may have undergone stochastic changes during passages. To avoid misunderstanding, we change the term "atypical" to "uncharacterized" in the revised manuscript.

We have previously shown that deletion of YKU70 does not affect MMEJ-mediated intra-chromosome fusion in single-chromosome SY14 cdc13Δ cells (Wu et al., 2020). In SY12 cells, double knockout of TLC1 and YKU resulted in synthetic lethality, and we were unable to continue our investigation. The result of synthetic lethality of TLC1 and YKU70 double deletion was shown in the Figure 7B in the reviewed preprint version 1, and the result was not included in the reviewed preprint version 2 in accordance with the reviewer's instructions.

"Atypical” survivors could be detected in SY13 cells (Figure 1D), but the frequency of their formation in the SY13 strain appeared to be lower than in SY12. As one can imagine, SY13 contains two chromosomes and its survivors should have a higher frequency of intra-chromosome fusions.

(2) From their data, it is possible that X and Y elements influence homologous recombination, type 1 and type 2 (type X), at telomeres. In particular, the presence of X and Y elements appears to be important for promoting type 1 recombination. In other words, although not essential, subtelomeres have some function in maintaining telomeres. I suggest that the authors include author response image 4 in the text. They could revise their conclusion and the paper title accordingly.

According to this suggestion, we have included author response image 4 in the revised manuscript as Figure 2E, Figure 5D, Figure 6C and Figure 6E. Accordingly, we have changed the title as “Elimination of subtelomeric repeat sequences exerts little effect on telomere essential functions in *Saccharomyces cerevisiae*”.

(3) Minor points: The newly added data indicate that X survivors are generated in a type 2-dependent manner. The authors could discuss how Y elements were eroded while retaining X elements (line 225, Figure 2A).

Thank this reviewer’s suggestion. We have discussed it in the revised manuscript (p.13 line 244-245). When telomere was deprotected, chromosome end resection took place. Since SY12 only has one Y’-element, it is hard to search homology sequences to repair the Y’-element in XVI-L. When the X-element in XVI-L was exposed by further resection, it is easier to find homology sequences to repair. So, in Type X survivor the Y’-element was eroded while retaining X-element.

**Reviewer #2**
I would like to congratulate the authors for their work and the efforts they put in improving the manuscript. The major criticism I had previously, ie testing the genetic requirements for the survivor subtypes, has been met. Below are a few minor comments that don't necessarily require a response.(1) I think the Author response image 6 could have been included in the manuscript. I understand that the authors don't want to overinterpret survivor subtype frequencies, but this figure would have suggested some implication of Rad51 in the emergence of survivors even in the absence of Y' elements. At this stage, however, it is up to the authors, and leaving this figure out is also fine in my opinion.

According to the suggestion, the author response image 6 has been presented as Figure 6—figure supplement 7.

(2) Chromosome circularization seems to rely on microhomologies. Previously, the authors proposed that SY14 circularization depended on SSA (Wu et al. 2020), but here, since circularization appears to be Rad52-independent, it is likely to be based on MMEJ rather than SSA (although there are contradictory results on Rad52's role in MMEJ in the literature).

Yes, we mentioned it in the revised manuscript.

(3) p. 28 lines 511-513: "The erosion sites and fusion sequences differed from those observed in SY12 tlc1Δ-C1 cells (Figure 2D), suggesting the stochastic nature of chromosomal circularization": I don't think they are necessarily stochastic, because the sequences beyond the telomeres are now modified, the available microhomologies have changed as well.

We agreed with your opinion. In different chromosomes, there tend to be some hotspots for chromosome fusion. For example, in Figure 6C and 6F the resection site in Chr1 and Chr2 was the same in SY12XYΔ+Y tlc1Δ-C1 and SY12XYΔ tlc1Δ-C1. So, we speculate that there are some hotspots for chromosome fusion, but which site the cell will choose in one round chromosome fusion event is stochastic.

(4) Typos and other errors:p. 3 line 52: "subtelomerice" and "varies" are mispelled.p. 5 line 78: "processes" should be "process".Supp files are mislabelled (the numbers do not correspond to file name).Supp file 2: how come SY12 has only one Y' element and SY13 has two?p. 10 line 175: "emerging" should be "emergence".p.15 line 276: "counter-selected" should be "being counter-selected" or "counterselection".p. 29 line 523: "the formation of them" should be "their formation".p. 37 line 653: "could have been an ideal tool": the sentence is grammatically incorrect. Writing "AND could have been an ideal tool" is enough to make it structurally correct.

Thanks for pointing these errors out. We have corrected them in the revised manuscript. For the question “how come SY12 has only one Y' element and SY13 has two?” we were not sure at this moment. We speculated that one of the Y’ might be lost during genetic engineering of the chromosomes by CRISPR–Cas9 system.

**Reviewer #3**
The authors included statistical analyses of the qPCR data (Fig 4B) as requested, but did not comment on the striking difference in expression of MPH3 and HSP32 in the SY12 strain compared to BY4742. An improvement of the manuscript is the inclusion of rad52 tlc1 strains in their analyses, demonstrating that the "atypical and circular survivors" arose independently of homologous recombination. In addition, by analyzing rad51 and rad50 mutant strain they could demonstrate that the "type X" survivors had similar molecular requirements to type II survivors. Overall, the revised submission improves the article.

We thank the reviewer’s comments and suggestions. The SY12 strain (with three chromosomes) exhibited lower expression levels of both MPH3 and HSP32 compared to the parental strain BY4742 (with 16 chromosomes). We speculated that with the reduced chromosome numbers, the silencing proteins appeared to no longer be titrated by other telomeres that have been deleted. We have added these comments in the revised manuscript.

Wu, Z.J., Liu, J.C., Man, X., Gu, X., Li, T.Y., Cai, C., He, M.H., Shao, Y., Lu, N., Xue, X., et al. (2020). Cdc13 is predominant over Stn1 and Ten1 in preventing chromosome end fusions. Elife 9.